# Evaluating the transmission dynamics and host competency of aoudad (*Ammotragus lervia*) experimentally infected with *Mycoplasma ovipneumoniae* and leukotoxigenic *Pasteurellaceae*

**Logan F. Thomas**[1]*, **Dallas Clontz**[1], **Chase M. Nunez**[1], **Robert O. Dittmar**[2†], **Froylán Hernandez**[2], **Raquel R. Rech**[1], **Walter E. Cook**[1]

**1** Department of Veterinary Pathobiology, School of Veterinary Medicine and Biomedical Sciences, College of Agriculture and Life Sciences, Texas A&M University, College Station, Texas, United States of America, **2** Big Game Program, Texas Parks and Wildlife Department, Austin, Texas, United States of America

† Deceased.
* Logan201@tamu.edu

## Abstract

Feral populations of aoudad (*Ammotragus lervia*) occur in Texas bighorn sheep (*Ovis canadensis*) habitat and pose several conceptual ecological threats to bighorn sheep re-establishment efforts. The potential threat of disease transmission from aoudad to bighorn sheep may exacerbate these issues, but the host competency of aoudad and subsequent pathophysiology and transmissibility of pneumonic pathogens involved in the bighorn sheep respiratory disease complex is largely unknown. Because the largest population-limiting diseases of bighorn sheep involve pathogens causing bronchopneumonia, we evaluated the host competency of aoudad for *Mycoplasma ovipneumoniae* and leukotoxigenic *Pasteurellaceae*. Specifically, we described the shedding dynamics, pathogen carriage, seroconversion, clinical patterns, and pathological effects of experimental infection among wild aoudad held in captivity. We found that aoudad are competent hosts capable of maintaining and intraspecifically transmitting *Mycoplasma ovipneumoniae* and *Pasteurellaceae* and can shed the bacteria for 53 days after exposure. Aoudad developed limited clinical signs and pathological findings ranged from mild chronic lymphohistiocytic bronchointerstitial pneumonia to severe and acute suppurative pneumonia, similarly, observed in bighorn sheep infected with *Mycoplasma* spp. and Pasteurellaceae bacteria, respectively. Furthermore, as expected, clinical signs and lesions were often more severe in aoudad inoculated with a combination of *Mycoplasma ovipneumoniae* and *Pasteurellaceae* as compared to aoudad inoculated with only *Mycoplasma ovipneumoniae*. There may be evidence of interindividual susceptibility, pathogenicity, and/or transmissibility, indicated by individual aoudad maintaining varying severities of chronic infection who may be carriers continuously shedding pathogens. This is the first study to date to demonstrate that aoudad are a conceptual disease transmission threat to sympatric bighorn

**Data Availability Statement:** All relevant data are within the paper and its Supporting Information files.

**Funding:** This study was funded by the Foundation for North American Wild Sheep, Grant-In-Aid Project Number "1920-71", (Walter E. Cook.) The funder had no role in study design, data collection and analysis, decision to publish, or preparation of the manuscript.

**Competing interests:** The authors have declared that no competing interests exist.

sheep populations due to their host competency and intraspecific transmission capabilities.

## Introduction

Nonindigenous species are of utmost importance to modern wildlife management, as they can harm native species through actions like direct resource competition [1], reconstruction of food web and trophic interactions [2], and alteration of disease transmission dynamics [3, 4]. As such, species introductions are a major contributing factor to global biodiversity loss [5]. The state of Texas' history of ungulate game introductions provides unique opportunities to study interactions between exotic and native species. Captive commingling studies demonstrated that the highly adaptable white-tailed deer (*Odocoileus virginianus*) experienced decreased growth, reproduction, and increased local extirpation risk in the presence of exotic ungulates [6]. One of these deleterious species was the aoudad/Barbary sheep (*Ammotragus lervia*).

In the late 1950s, aoudad (AD) were introduced from the Barbary Coast of northern Africa to the southwestern United States where they established feral populations [7]. Texas statewide population estimates were 3,750 in the 1980s and upwards of 20,000 by 1999 [7, 8]. There are currently over 8,000 AD occupying native desert bighorn sheep (*Ovis canadensis nelsoni*, BHS) habitat in the Trans-Pecos region of Texas [9, 10]. AD have greater diet plasticity [11–13] than BHS, are more fecund [14], and utilize similar habitats [15]. For these reasons, wildlife managers are concerned AD may hinder BHS population recovery/re-establishment [10, 16]. Recent serologic evidence of novel exposure of AD and BHS in the Trans-Pecos region of Texas to a population-limiting respiratory pathogen, *Mycoplasma ovipneumoniae*, elevates these conservation concerns [17, 18].

Notably, AD were also introduced to Europe and they are currently expanding their range and ecological impact in countries like Spain [19]. A similar disease concern exists between AD and native Spanish ibex (*Capra pyrenaica hispanica*). Several ibex species and AD are both susceptible to population-limiting disease due to sarcoptic mange caused by the highly transmissible *Sarcoptes scabiei* mite [20, 21]. Thus, AD disease ecology is an international issue deserving more holistic attention across their native and introduced ranges.

Bronchopneumonia is considered a population limiting disease among bighorn sheep that can cause all-age die-offs and subsequently hinder lamb recruitment for several years after an outbreak, occasionally eliminating entire BHS herds [17, 18]. *Mycoplasma ovipneumoniae* is a primary causative agent of polymicrobial bronchopneumonia in BHS populations across the United States and Canada [18, 22, 23]. Healthy BHS can contract *M. ovipneumoniae* from infected domestic goats [24, 25] (*Capra aegagrus hircus*), other BHS, mountain goats [26] (*Oreamnos americanus*), and domestic sheep [27] (DS; *Ovis aries*), but the potential for AD to serve as a reservoir for *M. ovipneumoniae* and other BHS respiratory pathogens is unknown. The evolutionary relationships of known disease reservoir species to BHS suggests that host competency for *M. ovipneumoniae* may be highly conserved in the *Caprinae* subfamily to which aoudad belong [28]. Additionally, the host range and prevalence of *M. ovipneumoniae* is likely underestimated in wild ruminants, which raises many questions about bronchopneumonia epidemiology where multiple potential hosts occur [29]. In addition to *M. ovipneumoniae*, several *Pasteurellaceae* species have been implicated in severe BHS bronchopneumonia epizootics and are known to act as synergistic pathogens during and/or after *M. ovipneumoniae* infection [23, 30]. Serology, culture, and molecular data from tissues (2010–2018) from the

Texas Parks and Wildlife Department (TPWD) disease surveillance program had not recorded pneumonia epizootics or *M. ovipneumoniae* in BHS. For these reasons, BHS in Texas are thought to be largely free of circulating *M. ovipneumoniae* and to be of low pneumonia risk. The present study assesses the risk AD may pose to BHS in Texas by determining whether AD can carry and transmit *M. ovipneumoniae* and leukotoxigenic *Pasteurellaceae*.

We evaluated the host competency of AD to *M. ovipneumoniae*, with and without co-exposure to DS lung flora, including leukotoxigenic strains of *Pasteurellaceae*, and their ability to cause clinical disease in this species. We also evaluated the shedding dynamics of experimentally infected AD to determine AD transmission propensity and describe the pneumonia patterns of AD exposed to these agents. Addressing these objectives will help inform wildlife managers on the relative transmission risk infected AD may pose to BHS and subsequently inform future research and management decisions involving these species.

## Materials and methods

### Animals and housing

All activities and procedures were approved by Texas A&M University's Institutional Animal Care and Use Committee under Animal Use Permit 2019–0296. We captured 21 (eleven female, ten male) AD in the Carrizo Mountain range approximately 10 miles south of the Van Horn, Texas (31˚00'46.0"N 104˚55'51.8"W). Prior to inoculation procedures, we experienced 7 mortalities: 2 perianesthetic mortalities (AD #17; AD #18), 1 mortality due to haemonchosis (AD# 11; *Haemonchus* sp.), 1 mortality due to pregnancy toxemia (AD #2), and 3 euthanasia events (AD #10; AD #19; AD #20). AD #10 was euthanized due to poor body condition, AD #19 was euthanized due to lameness, and AD #20 was euthanized due to a traumatic injury experienced during off-loading animals from the trailer. As a result, we used 14 AD in our study (AD #1, AD #3–9, AD #12–16, and AD #21). Further details on chemical immobilization procedures and euthanasia protocols are published [31]. Animals varied from sub-yearling to approximately five years of age indicated by tooth eruption and horn characteristics [32, 33]. AD were ear-tagged and randomly assigned to one of three groups based on the following post-acclimatization inoculation schemes: 1) inoculation with isolated *M. ovipneumoniae* (Movi group; n = 6), 2) inoculation with nasal washings from *M. ovipneumoniae*-positive DS (Wash group; n = 5), or 3) no inoculation with fence-line contacts allowed with the Movi and Wash groups (control-contact group; n = 3). The AD were then transported non-stop in a single trailer physically separated by treatment groups to the study site at Veterinary Medical Research Park at Texas A&M University in College Station, Texas (30˚36'40.5"N, 96˚21'57.5"W). AD were then placed in three separate pens (>500 m$^2$ each) corresponding to their treatment group and afforded 24 (Movi group) to 38 days (Wash group) to acclimatize to captivity (acclimatization period). Animals had unlimited access to pasture, hay, and water and were provided supplemental protein pellet feed for the duration of the study. Body condition was estimated daily by visual inspection and at each capture event by manual palpation. We used a daily clinical scoring system to evaluate pneumonia pathogenesis, potential for pathogen shedding, and animal welfare throughout the study [24]. Specifically, criteria for euthanasia was indicated by a score of 5 or higher for 5 consecutive days during the 86-day study period.

### Animal restraint and handling

Capture in the Carrizo mountains was by netgun delivered via helicopter with leg hobbles and blindfolds placed for physical restraint. Immediately upon capture, all animals were given 15 mg azaperone hand-injected intramuscularly and given 0.2–0.7 mg/kg sustained release

haloperidol (ZooPharm, Laramie, Wyoming 82070, USA) 30–45 mins after capture to alleviate the stress of capture, translocation, and to aid acclimatization to captivity [31, 34].

To collect biological samples, AD were chemically immobilized via dart projector (dart gun) with a combination of nalbuphine (40mg/mL; 0.7 to 2.4 mg/kg), azaperone (10mg/mL; 0.26 to 0.43 mg/kg), and medetomidine (10mg/mL; 0.26 to 0.43 mg/kg) [31]. Stress was managed by expeditiously darting all animals in the group, rigorously monitoring vital signs, and minimizing the period animals were immobilized. Further details on capture events are published [31]. After inoculation, the Movi group (n = 6) was sampled approximately weekly seven times while the Wash group (n = 5) was sampled approximately weekly five times after their respective inoculation date (38 days post-arrival). A medetomidine (5 mg/mL; 0.08 to 0.25 mg/kg) plus ketamine (150 mg/mL; 2.4 to7.5 mg/kg) combination (MK, ZooPharm, Laramie, Wyoming 82070, USA) was used in the two sampling events of control-contact-contact group AD (n = 3). After sampling upon initial capture, control-contact group AD were only captured and sampled immediately prior to euthanasia 86 days post-arrival. Per veterinary recommendation from the Veterinary Medical Research Park staff, all AD were administered 0.2 mg/kg ivermectin (vetrimec 1%, 10 mg/mL, VetOne, Boise, Idaho 83705) on day 45 of the study. Prior to this, no clinical indication of parasitism was observed, and treatment was only applied once body condition(s) began to decline to minimize the risk of developing ivermectin resistance. Animals were housed in groups to avoid isolation stress and personnel activity was minimized around their housing facilities.

## Inoculation

The Movi group inoculum was created using six quantitative polymerase chain reaction (qPCR) -positive [35–37] nasal swabs from a naturally infected BHS at the Tom Thorne and Beth Williams Wildlife Research Center at Sybille's captive herd. This individual consistently and chronically sheds *M. ovipneumoniae* and nasal swab samples from this individual are routinely used by Wyoming State Veterinary Laboratory (WSVL) as a source of positive control samples. Swab samples from this individual were confirmed as culture positive by WSVL. Swabs were cultured in tryptic soy broth (TSB; 0.03 g/mL tryptic soy broth powder, 15% glycerol, 85% deionized water) modified with 7.24 μg/mL amphotericin B and 0.88 mg/mL penicillin G potassium salt to exclude the growth of cell-wall producing bacterial species. The modified TSB was incubated at 37°C for approximately 48 hours at 10% carbon dioxide then stored at -70°C. Samples were pooled and diluted with 175 mL of sterile phosphate-buffered saline (PBS). Inoculum for the Wash group was obtained by nasal PBS irrigation of four *M. ovipneumoniae*-positive DS verified by Washington Animal Disease Diagnostic Laboratory's (WADDL) qPCR two weeks prior to irrigation [35, 38]. Due to the consistent shedding of *M. ovipneumoniae* in exposed DS, source animals were only tested for *M. ovipneumoniae* shedding via qPCR once before collecting washes. Four of the five AD in the Wash group were inoculated with washes from a single DS. However, the fifth AD (AD #13) was inoculated with equal volumes of washings from all four qPCR-positive DS to yield a total of 25 mL of inoculum. This was done to examine the possibility of multi-strain *M. ovipneumoniae* infection and its potential role in pathogenesis and transmission propensity. Although we did not confirm the presence of *M. haemolytica* in the Wash group inoculum prior to administration, this agent is a frequent constituent in the domestic sheep respiratory tract and most strains are leukotoxigenic [30].

Following the precedent of Dr. Thomas Besser's work in *M. ovipneumoniae* experimental inoculations, all AD of the Movi and Wash groups were inoculated by instilling 25 mL of each respective inoculum into the nares (5 mL each), pharynx (10 mL), and conjunctival sacs (2.5

mL each) via 30 cc irrigation syringe [24]. Inoculation occurred immediately after sampling AD at the end of the acclimatization periods (Movi group, 24 days post-arrival; Wash group, 38 days post-arrival). Any AD testing qPCR negative upon nasal swab sampling and exhibiting no clinical signs at seven days post-inoculation (p.i) were re-inoculated as described above. We were unable to determine the dose or concentration of *M. ovipneumoniae* in either inoculum due to the inherent difficulty in quantifying these species [39]. Further, the focus of our study was on pathogen carrying potential of AD rather than attempting to simulate natural transmission with our inoculation procedures.

## Pathogen detection

Upon initial capture, blood was collected via jugular venipuncture for complete blood count (CBC) analyses and to detect antibodies to bovine herpesvirus type 1 (BHV-1), bovine viral diarrhea types 1a/1b/2 (BVDV), parainfluenza 3 virus (PIV3), and bovine respiratory syncytial virus (BRSV) using Texas Veterinary Medical Diagnostic Laboratory's (TVMDL) bovine respiratory serology panel. After capture in the Carrizos mountains, CBCs were also performed on blood samples collected upon inoculation and all p.i sampling events. Antibodies to *M. ovipneumoniae* were detected by competitive enzyme-linked immunoabsorbent assay (cELISA) at WADDL from all animals upon initial capture in the Carrizos mountains and at each sampling event starting at 21- and 14-days p.i for the Movi and Wash groups, respectively.

To assess the bacterial composition of the upper respiratory tract, with a focus on the presence of *Pasteurellaceae*, nasopharyngeal tonsil swabs were collected with sterile rayon-flocked swabs and placed in Ames gel media with charcoal (Becton Dickson, Franklin Lakes, NJ 07417, USA) and cultured under both aerobic and anaerobic conditions at TVMDL on cation adjusted Mueller-Hinton agar for 24–48 hours. After culturing, colonies were isolated and removed by experienced diagnostic microbiologists and speciated by matrix assisted laser desorption ionization-time of flight mass spectrometry (MALDI) at TVMDL. Nasal swabs for detecting *M. ovipneumoniae* by qPCR at WADDL were collected with sterile polyester-tipped swabs (Puritan, Guilford, ME 04967, USA) and stored in two mL of TSB and frozen immediately at -20˚C. Nasal swabs were classified based on cycle threshold (CT) values as follows: $\geq$40 negative, <36 positive, 36–40 indeterminate [35–38]. To investigate the occurrence of *M. ovipneumoniae* transmission in inoculated AD groups, we employed multi-locus Sanger sequencing (MLST) at WADDL [40] on nasal washings from all DS individuals and isolated *M. ovipneumoniae* from BHS nasal swabs. The aim was to identify strains of *M. ovipneumoniae* in nasal swabs that were absent in the initial inoculum for each individual. Detection of such (a) strain(s) in the nasal swabs would indicate natural transmission events. For the control-contact group, detection of *M. ovipneumoniae* by qPCR was considered suggestive of transmission.

Analysis of *M. ovipneumoniae* status of nasal swabs upon initial capture, however, was achieved by full-length next-generation sequencing of the 16S ribosomal subunit performed with a proprietary protocol by MicroGen Diagnostics (Lubbock, Texas 79407, USA). This was done in lieu of testing nasal swabs by qPCR for a study investigating microbiome changes in response to capture and translocation of AD to the study site.

Upon euthanasia (11/14 aoudad via intravenous pentobarbital sodium 85.8 mg/kg; 390 mg/mL) or spontaneous mortality (3/14 aoudad), necropsies and histopathological analyses of 10% formalin-fixed tissues were conducted. Sections of lung representative of each lung lobe and tracheobronchial lymph nodes were held at -80˚C and submitted for *Pasteurellaceae* isolation by culture, detection of the leukotoxin-A gene from *Mannheimia haemolytica* and/or *Bibersteinia trehalosi* via PCR, and *M. ovipneumoniae* detection via qPCR at WADDL. Lung sections and tracheobronchial lymph nodes were also cultured on both Columbia blood agar

(5%) under 5% carbon dioxide conditions and MacConkey agar with no gas modifications at 35˚C both for 24–72 hours.

## Statistical analyses

All statistics were performed with program R, version 4.2.2. Sampling days with only one group (Wash, Movi, or control-contact) sampled were matched with the closest alternative sampling day(s) to provide the closest p.i comparisons between the Movi, Wash, and control-contact groups. We compared the frequencies of *M. ovipneumoniae* detection, *Pasteurellaceae* species detection, and *M. ovipneumoniae*-specific antibody detection using Fisher's exact test with Holm's adjustment. We compared the daily cumulative mean clinical scores (CMS) between the Movi, Wash, and control-contact groups throughout the study using the unpaired two-sample Wilcoxon's test with Holm's adjustment. CMS for each treatment group for each day was determined by calculating the average of each treatment group's distribution of individual daily cumulative mean clinical scores (one clinical score per animal per day). That is, the cumulative mean clinical score of each AD was calculated each day and the mean of these values was calculated for each treatment group for a given day. Clinical scores and CMS were compared with the unpaired two-sample Wilcoxon's test with Holm's adjustment across days p.i only for animals which developed visibly apparent clinical signs after inoculation or potential interactions with inoculated individuals. Wilcoxon rank sum tests were performed to identify significant changes in each blood parameter in pre- versus p.i samples and detect differences between groups both study-wide and at p.i timepoints. We report the ratio of median absolute deviance to median (MADM) to evaluate the relative variation in each blood parameter and clinical score metric [41]. Spearman's rank correlation test was used to identify temporal associations between clinical score metrics and CBC values. All hypothesis tests and tests of associations were two-tailed with a significance level of $\alpha \leq 0.05$.

## Results

### Clinical condition, mortality, and euthanasia

We experienced 3 p.i mortality events prior to euthanasia; Movi group AD #6, Wash group AD #9, and Wash group AD #14. These animals died 21-, 49-, and 8- days p.i, respectively. AD #6 died during sampling under chemical immobilization. AD #9 died of haemonchosis (*Haemonchus* sp.), and AD #14 died of pneumonia (described later). Movi group AD #1 was euthanized 53-days p.i and the rest were euthanized 61-days p.i (AD # 3–7). Wash group AD #13 was euthanized 42days p.i and the rest were euthanized 46-days p.i (AD #8; AD #12). AD #1 and AD #13 were euthanized prior to their respective penmates due to a progressive decrease in body condition. Control-contact AD were all euthanized 86-days post-arrival. Euthanasia dates were unbalanced and variable due to challenges recruiting sufficient personnel for initial inoculations (yielding different acclimatization times between Movi and Wash groups) and, later, limitations in pathologist capacity (limited numbers of carcasses could be processed on the same day).

The mean clinical score among inoculated AD was 0.26 (SD = 0.30) when days p.i were standardized between Movi and Wash groups. The study-wide mean clinical score among control-contact AD was 0.0171 (SD = 0.2065) and the median CMS for this group was 0 (MADM = 0). The median CMS among inoculated AD was 0.177 (MADM = 0.78). Final median CMS for Movi group (0.17, MADM = 0.64) and Wash group AD (0.18, MADM = 1.04) were similar (W = 11, p = 0.84) (Fig 1). Wash group AD had significantly greater CMS than Movi group AD each day between 10- and 13- days p.i (W = 1.25, p = 0.03). Wash group AD also developed clinical signs more rapidly ($X^2$ = 5.4, p = 0.02) and reached a

## Clinical Score Comparison

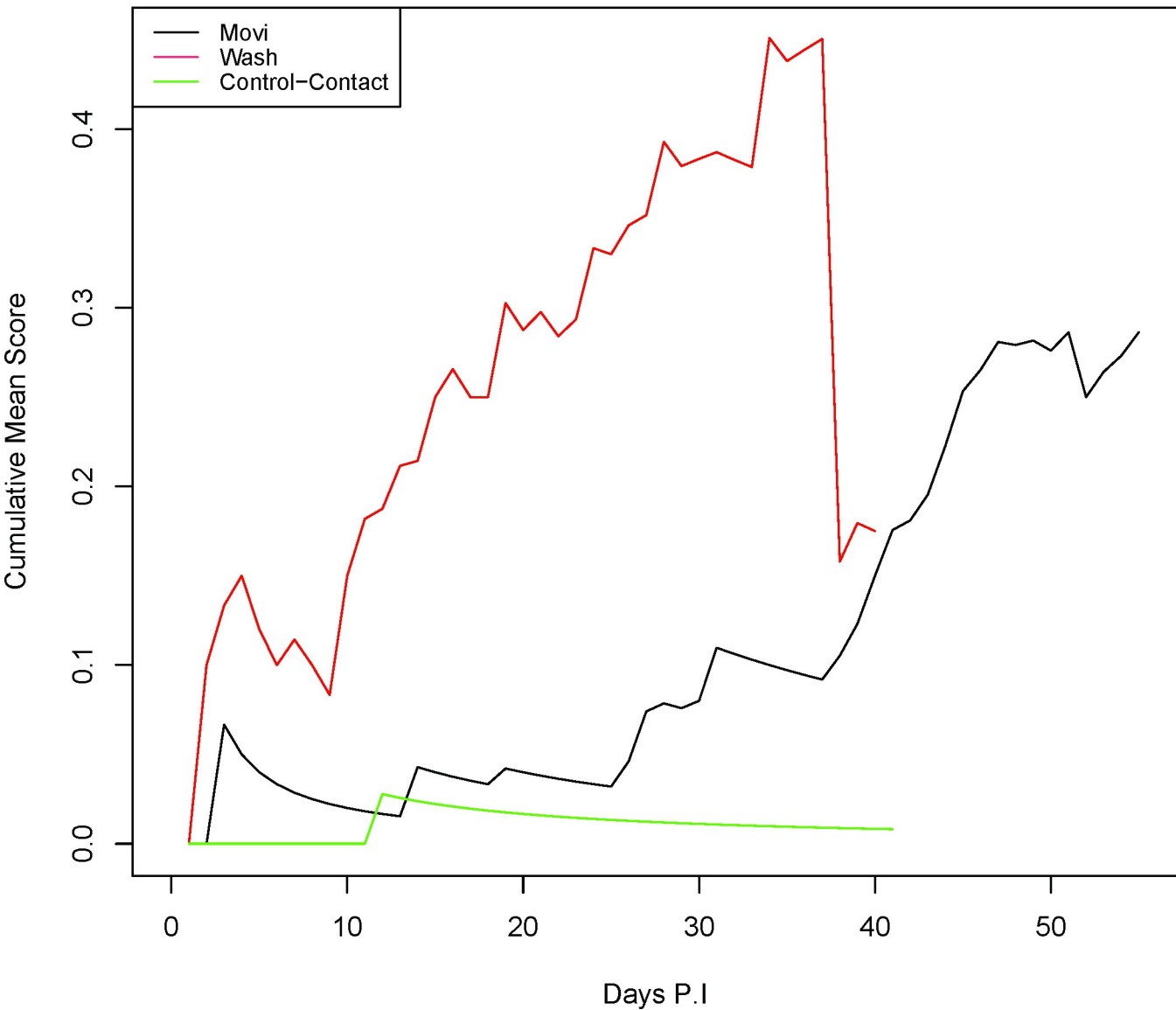

**Fig 1. Clinical disease severity observed in aoudad (*Ammotragus lervia*) experimentally infected with *Mycoplasma ovipneumoniae* (Movi group) or nasal lavage fluid from *M. ovipneumoniae* PCR positive domestic sheep (Wash group).** Cumulative mean clinical score compared longitudinally between Movi and Wash aoudad treatment groups. The acute decrease in Wash group CMS occurred due to a mortality event of an individual with consistent clinical signs.

clinical score of at least two earlier ($X^2$ = 55, p = 0.03). The decrease in CMS for the Wash group at approximately 38 days p.i was due to euthanasia of AD #13.

### Mycoplasma ovipneumoniae

Six Movi and five Wash group AD were inoculated, and three control-contact AD shared a fenceline (allowed nose-to-nose contact) with these two inoculated groups. This afforded transmission opportunities between control-contact AD and both inoculated AD groups. We collected 80 nasal swabs, 61% (49/80) of which tested qPCR positive, 16% (13/80) tested

**Table 1. Proportion of nasal swabs from aoudad (*Ammotragus lervia*) testing positive for *Mycoplasma ovipnuemoniae* by qPCR.**

| Treatment | Capture ‡ | Days P.I. | | | | | | | | P.I Total | Study-wide Total |
|---|---|---|---|---|---|---|---|---|---|---|---|
| | | 0 | 7 | 14 | 21 Movi 20 Wash | 28 Movi 31 Wash | 34 Movi 39 Wash | 45 | 53 | | |
| Movi | 0/2 (4 Ins) | 6/6 | 5/6 (*1) | NA | 6/6 | 5/5 | 4/5 (*1) | 4/5 (*1) | 4/5 | 28/32 (*3) | 34/44 (*3) |
| Wash | 0/3 (2 Ins) | 1/5 | 3/5 (*1) | 2/4 (*2) | 4/4 | 3/4 (*1) | 2/3 (*1) | NA | NA | 14/20 (*5) | 15/30 (*5) |
| Control-contact | 0/3 | | | | | | | | | 1/3 (*1) | 1/6 (*1) |

Proportions of aoudad nasal swabs testing qPCR positive; Days P.I, days post-inoculation; (Ins), insufficient sample due to PCR inhibitors (*), indicates the number of samples testing indeterminate (qPCR cycle threshold value 36–40); Total P.I, aggregate of all samples collected after each respective treatment group was inoculated; ‡, indicates samples were analyzed by next-generation sequencing of the 16S ribosomal subunit at MicroGen Dx rather than qPCR at Washington Animal Disease Diagnostic Laboratory; NA, indicates that animals were not sampled at a given sampling day/event.

indeterminate, and 14% (11/80) tested negative. The nasal swabs collected upon capture in the Carrizos mountains, however, were subjected to full-length next-generation sequencing of the 16S ribosomal subunit to detect *M. ovipneumoniae* rather than qPCR at WADDL. Several nasal swabs (n = 7) collected upon capture in the Carrizos mountains were insufficient due to poor amplification and/or PCR inhibitors [42]. These comprised 9% of the total nasal swabs collected. *M. ovipneumoniae* was detected by qPCR consistently in the Movi and Wash groups after inoculation, and in the control-contact AD upon study termination (Table 1). Samples from all three AD groups yielded indeterminate qPCR results throughout the study with no apparent or statistical relationship with group identity or study timepoint.

*M. ovipneumoniae* was detected by qPCR in all Movi group (6/6) AD nasal swabs collected immediately pre-inoculation. Seven days p.i, 17% (1/6) Movi group AD (AD#1) tested positive on initial testing with qPCR. Accordingly, the Movi group was re-inoculated four weeks after initial inoculation to verify animals were colonized. After subjecting nasal swabs collected seven days p.i to a second iteration of qPCR at WADDL, however, *M. ovipneumoniae* was detected in 83% (5/6) of Movi group samples. All samples were handled equivalently upon collection and storage, and no known cause was identified for the diagnostic discrepancy between the first and second iteration of Movi group nasal swab samples subjected to qPCR at WADDL. Because Movi group AD were apparently colonized at seven days p.i, we used the day these individuals were first inoculated (day 0, 24 days post-arrival) as a reference for our clinical and pathogen detection comparisons throughout the study. The Wash group was inoculated (day 0) 38 days post-arrival when *M. ovipneumoniae* was detected by qPCR in one (1/5) Wash group AD. *M. ovipneumoniae* detection rates did not differ between treatment groups p.i (OR = 2.9 to 12.3, p = 0.15 to 0.069). Likewise, *M. ovipneumoniae* detection was similar between inoculated groups at all matched p.i timepoints.

When DS nasal washes (samples 19391, 19393, 19377, and 19379) and isolated *M. ovipneumoniae* (sample C32) were subjected to MLST at WADDL, samples C32, 19391, 19393, 19377, and 19379 amplified zero, two, one, four, and zero of the four targeted loci. Sample C32 was culture and qPCR positive at WSVL but yielded high CT values corresponding to low-positive (35.41) and indeterminate (36.71) results. Seven days prior to collecting nasal washes, a nasal swab collected from the DS providing nasal wash sample 19379 tested qPCR at WADDL with a relatively low CT of 29.336. Further details on MLST are available (S1 Appendix).

After discovering seropositive AD in the Movi (AD #2) and Wash (AD #9) groups pre-arrival, we subjected pre-inoculation nasal swabs from these individuals to MLST to distinguish any natural/pre-existing strains from those present in the inoculum for either group. Due to AD #13's rapid clinical onset (possibly indicative of pre-inoculation *M. ovipneumoniae* infection), we also subjected this individual's pre-inoculation nasal swab to MLST. However,

**Table 2. Proportion of aoudad (*Ammotragus lervia*) serum samples testing positive for *Mycoplasma ovipneumoniae* antibodies by cELISA.**

| Treatment | Days P.I | | | | | | | | | P.I Total | Study-wide Total |
|---|---|---|---|---|---|---|---|---|---|---|---|
| | -24 | -38 | 14 | 21 Movi 20 Wash | 28 Movi 31 Wash | 34 Movi 39 Wash | 45 Movi 41 Wash | 53 Movi | | | |
| Movi | 1/6 | NA | NA | 3/6 | 3/6 (*1) | 5/5 | 5/5 | 5/5 | | 21/27 (*1) | 22/33 (*1) |
| Wash | NA | 1/5 | 2/4 (*2) | 4/4 | 4/4 | 1/3 (*1) | 3/3 | NA | | 14/18 (*3) | 15/23 (*3) |
| Control-contact | 0/3 | | | | | | | | | 2/3 | 2/6 |

Proportion of aoudad testing cELISA positive for antibodies to *Mycoplasma ovipneumoniae* at Washington Animal Disease Diagnostic Laboratory; Days P.I, days post-inoculation

*, indicates the number of samples testing indeterminate (40%-50% inhibition); NA, indicates that animals were not sampled at a given sampling day/event.

no amplification was observed in nasal swabs from AD #2 or #9. Amplification was observed for all four loci from AD #13, but the sequences for all our loci did not converge to allow strain-typing.

After the respective inoculation dates for the Movi and Wash groups, AD in both groups were consistently seropositive throughout the study and we observed seroconversion in control-contact AD upon study termination (Table 2). However, indeterminate cELISA results were observed in all three AD groups regardless of study timepoint. Study-wide p.i seropositivity rates in the Movi group (OR = 2.63, p = 0.44) and Wash group (OR = 1.93, p = 0.54) were like one another and the control-contact-contact group. Additionally, no significant differences were found for seropositivity rates at any paired timepoints throughout the study.

## *Pasteurellaceae* dynamics

Overall, *Pasteurellaceae* bacteria isolation (with subsequent species identification by MALDI) was significantly higher p.i (56%, 31/55) than pre-inoculation (20%, 5/25) (OR = 0.2, p = 0.0033). While isolation did not increase significantly for any single *Pasteurellaceae* species pre vs p.i, each *Pasteurellaceae* species was more frequently isolated p.i versus pre-inoculation (Table 3). However, the rate of "mixed" culture results that did not allow species discrimination was significantly higher pre-inoculation (68%, 17/25) than p.i (29%, 16/55) (OR = 5.06, p = 0.00146). Mixed culture results occurred when colonies could not be sufficiently separated for isolation and/or when colonies could not be properly isolated for species confirmation by MALDI at TVMDL. This can occur due to bacterial or fungal overgrowth that hinders isolation efforts. *T. pyogenes* demonstrated a modest (insignificant) increase in p.i isolation (OR = 0.33, p = 0.072) compared to pre-inoculation samples.

Compared to day zero results, *Pasteurellaceae* isolation frequency was significantly higher at 31- (100%, 4/4) (OR = Inf, p< 0.001) for the Wash group and 45- (100%, 5/5) (OR = Inf, p< 0.001) days p.i for the Movi group. *B. trehalosi* was detected more frequently at 45- (100%, 5/5) versus zero- (0%, 0/12) and seven- (0%, 0/11) days p.i (OR = Inf, p< 0.001).

*Pasteurellaceae* family isolation was similar between the Movi and Wash groups both study-wide and p.i, but the Wash group carried *M. haemolytica* more frequently in both study-wide (OR = 0.14, p = 0.004) and p.i samples (OR = 0.03, p = 0.001) compared to the Movi group.

*M. haemolytica* was exclusively isolated in the lungs Wash group AD #8, 9, and 14, but only AD # 8 and #9 tested qPCR positive for the leukotoxin-A gene at WADDL. *Pasteurella multocida* was isolated in the lungs of Movi group AD #1, Wash group AD #13, and control-contact AD #21. *B. trehalosi* was isolated in Movi group AD #4 and #7, as well as control-contact AD #15. Additional details on other bacteria cultured from lung samples are available (S1 Table).

**Table 3. Detection of *Pasteurellaceae* bacteria from aoudad (*Ammotragus lervia*) tonsil swabs by culture.**

| Treatment | Species | -24 Movi -38 Wash | 0 | 7 | 14 | Movi 21 Wash 20 | Movi 28 Wash 31 | Movi 34 Wash 39 | 45 | 53 | P.I Total (%) | Study wide Total (%) |
|---|---|---|---|---|---|---|---|---|---|---|---|---|
| | | | | | | | | **Days P.I** | | | | |
| **Movi** | **B. treh** | 2/6 | 0/6 | 0/6 | NA | 0/6 | 0/5 | 3/5 | 5/5 | 3/5 | 11/32 (34) | 13/44 (30) |
| | **M. haem** | 2/6 | 0/6 | 0/6 | | 0/6 | 0/5 | 1/5 | 0/5 | 0/5 | 1/32 (3) | 3/44 (7) |
| | **P. mult** | 0/6 | 0/6 | 1/6 | | 1/6 | 1/5 | 0/5 | 1/5 | 1/5 | 11/32 (16) | 5/44 (11) |
| | **T. pyo** | 0/6 | 0/6 | 0/6 | | 0/6 | 1/5 | 4/5 | 4/5 | 5/5 | 14/32 (44) | 15/44 (34) |
| | **Mixed** | 2/6 | 6/6 | 5/6 | | 3/6 | 3/5 | 1/5 | 0/5 | 0/5 | 12/32 (38) | 20/44 (45) |
| **Wash** | **B. treh** | 0/5 | 0/5 | 0/5 | 0/4 | 2/4 | 2/4 | 0/3 | NA | NA | 4/20 (20) | 4/30 (13) |
| | **M. haem** | 0/5 | 0/5 | 3/5 | 2/4 | 2/4 | 3/4 | 0/3 | | | 10/20 (50) | 10/30 (33) |
| | **P. mult** | 0/5 | 0/5 | 0/5 | 2/4 | 0/4 | 0/4 | 0/3 | | | 2/20 (10) | 2/30 (7) |
| | **T. pyo** | 2/5 | 1/5 | 0/5 | 1/4 | 2/4 | 1/4 | 0/3 | | | 4/20 (20) | 7/30 (23) |
| | **Mixed** | 3/5 | 2/5 | 2/5 | 0/4 | 0/4 | 0/4 | 2/3 | | | 4/20 (20) | 9/30 (30) |
| **Control-contact** | **B. treh** | 1/3 | | | | NA | | | | | 3/3 (100) | 4/6 (67) |
| | **M. haem** | 0/3 | | | | | | | | | 0/3 (0) | 0/6 (0) |
| | **P. mult** | 0/3 | | | | | | | | | 0/3 (0) | 0/6 (0) |
| | **T.pyo** | 0/3 | | | | | | | | | 1/3 (33) | 1/6 (17) |
| | **Mixed** | 2/3 | | | | | | | | | 0/3 (0) | 2/6 (33) |

Percentage and proportion of samples from which bacteria were isolated from tonsil swabs; Days P.I, days post-inoculation; P.I only; aggregate of samples collected after inoculation; B. treh, *Bibersteinia trehalosi*; M. haem, *Mannheimia haemolytica*; P. mult, *Pasteurella multocida*; T. pyo, *Truperella pyogenes*; Mixed, too many bacteria present for speciation/ isolation.

## Blood parameters

Relative neutrophils (W = 215, p = 0.0013), lymphocytes (W = 667, p < 0.001), relative lymphocytes (W = 612, p = 0.006), platelets (W = 84, p< 0.001), and fibrinogen (W = 238, p = 0.002) were all elevated post versus pre-inoculation. Wash group AD had higher study-wide total leukocytes (W = 279, p = 0.0017), neutrophils (W = 238, p = 0.004), relative neutrophils (W = 271, p = 0.019), relative lymphocytes (W = 580, p = 0.018), and relative monocytes (W = 488, p = 0.0082) than Movi group AD (Table 4). Control-contact AD had lower study-wide relative monocytes (W = 123, p = 0.04) than Movi group AD.

After inoculation, Wash group AD had significantly greater study-wide total leukocytes (W = 115, p = 0.04), neutrophils (W = 115, p = 0.04), and relative neutrophils (W = 88, p = 0.004) than Movi group AD. Movi group AD had higher p.i relative lymphocytes than Wash group AD (W = 290, p = 0.003).

**Table 4. Summary of study-wide blood parameters of aoudad (*Ammotragus lervia*).**

| Treatment | Monocytes K/uL median (MADM) | Lymphocytes K/uL median (MADM) | Neutrophils K/uL median (MADM) | Total Leukocytes K/uL median (MADM) | Fibrinogen K/uL median (MADM) | Plasma Proteins mg/dL median (MADM) | Platelets K/uL median (MADM) |
|---|---|---|---|---|---|---|---|
| Movi | 0.2 (0.74) | 1.7 (0.35) | 4.55 (0.47) | 6.6 (0.35) | 400 (0.37) | 7 (0.08) | 655 (0.40) |
| Wash | 0.2 (0.74) | 1.8 (0.49) | 7.5 (0.47) | 11.14 (0.36) | 400 (0.37) | 7 (0.06) | 578 (0.41) |
| Control-contact | 0.15 (0.49) | 2.7 (0.66) | 5.05 (0.18) | 9.75 (0.29) | 400 (0.37) | 6.9 (0.04) | 482 (0.39) |
| Overall | 0.2 (0.74) | 1.8 (0.41) | 5.6 (0.50) | 8.39 (0.48) | 400 (0.37) | 6.95 (0.08) | 585 (0.36) |

MADM, ratio of median absolute deviance to the median

Wash group AD had significantly greater lymphocytes immediately prior to inoculation (day 0) (W = 0, p = 0.02) compared to Movi group AD. Wash group platelets (W = 0, p = 0.04) and plasma protein levels (W = 0, p = 0.04) were greater than those of Movi group AD at approximately 20 days p.i. At approximately, 40-days p.i, Movi group AD had higher lymphocytes than Wash group AD (W = 15, p = 0.032). Longitudinal blood parameters are graphically illustrated (S1 Fig).

Finally, we detected no antibodies specific to PIV3, BVDV, or BHV-1. However, antibodies to BRSV were detected in one Movi group (AD #2) and one Wash group (AD #14) AD tested upon capture.

## Pathological findings

Overall, gross lesions of Wash group AD were more severe than those of the Movi group and consisted of pleural effusion (20%, AD #14), fibrinous pleuritis (20%, AD #14) (Fig 2D), fibrous pleural adhesions (80%, AD #8, 9, 13, 14), cranioventral lung consolidation (80%, AD #8, 9, 13, 14) (Fig 2E), mucopurulent exudate in the trachea and mainstem bronchi (60%, AD #8, 13, 14), and pulmonary abscesses (40%, AD #8, 13) (Fig 2F). One AD in the Wash group (20%, AD #12) had no gross lesions of pneumonia. One AD (33%, AD #21) in the control-contact group exhibited only fibrous pleural adhesions. More details on pulmonary and non-pulmonary lesion patterns are described (S2 Appendix).

Histopathologic findings of the lungs revealed four distinct lesion patterns described below: 1) Subacute to chronic, lymphohistiocytic bronchointerstitial pneumonia with perivascular, bronchial, and bronchiolar-associated lymphoid tissue (BALT) hyperplasia (Fig 3A–3C); 2) Acute suppurative bronchopneumonia (Fig 3D and 3E) with oat cells (Fig 3F); 3) Superimposition of patterns 1 and 2 (Fig 3G) with and without oat cells, intralesional bacteria (Fig 3H), abscess formation (Fig 3I), and pleuritis; 4) Acute intravascular neutrophilia with congestion and multifocal fibrin deposition.

Pattern 1 was observed in 100% of the Movi group, 60% (AD #8, 13, 14) of the Wash group, and 100% of the control-contact group. One AD in the Wash group (20%, AD #9) had lesions consistent with pattern 2. Pattern 3 was observed in 17% (AD #1) of the Movi group. In the Wash group, 60% (AD #8, 13, 14) of the AD exhibited pattern 3, one of which 20%, (AD #14) included oat cells, indicating presence of leukotoxigenic bacteria. One AD in the Wash group (20%, AD #12) and one in the Movi group (17%, AD #4) exhibited lesions of pattern 4, and these were superimposed over pattern 1 lesions in the Movi AD.

## Discussion

Regardless of potential pre-capture exposure to these respiratory agents, our results suggest AD are competent hosts for *Mycoplasma ovipneumoniae* with or without coinfection with leukotoxigenic *Pasteurellaceae* species or other DS respiratory microbiome constituents. Consistent *M. ovipneumoniae* detection in inoculated AD suggests this species is a competent host that can harbor and shed *M. ovipneumoniae* for 53 days after exposure. Molecular and serologic evidence of *M. ovipneumoniae* exposure in control-contact AD found upon euthanasia suggests natural transmission occurred in the present study and that this species seroconverts after natural exposure/transmission. The observed serologic evidence of *M. ovipneumoniae* exposure in two AD (Movi group AD #2, and Wash group AD #9) found upon capture suggests this respiratory agent may already be naturally circulating in free-ranging Texas AD populations. However, we were unable to formally confirm the potential sources of *M. ovipneumoniae* exposures (natural versus inoculum) because our MLST results did not allow us to discriminate *M. ovipneumoniae* strains in either inoculum from any strain(s) present in

Movi group

Wash group

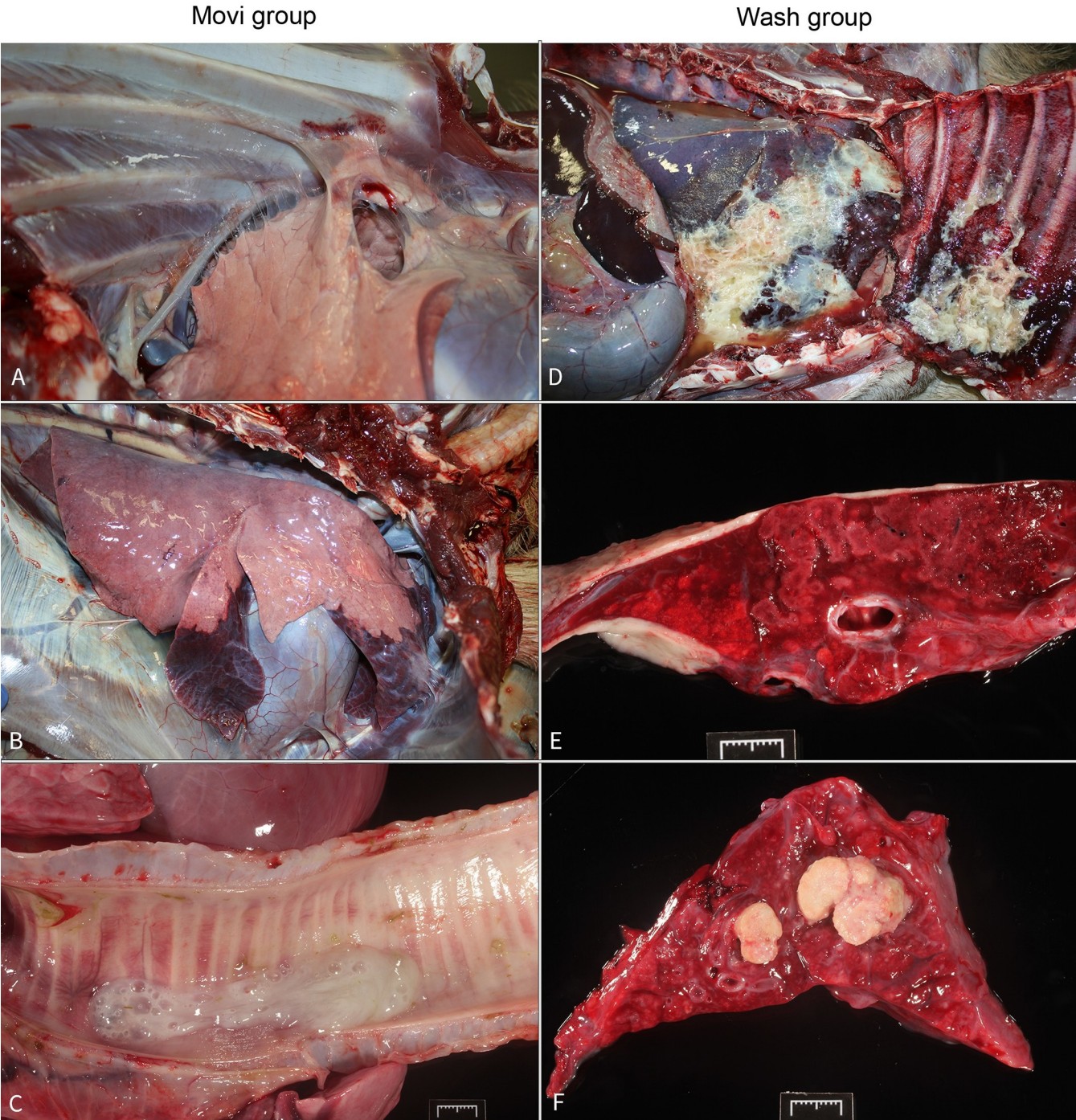

**Fig 2.** A-F. Pulmonary lesions of aoudad (*Ammotragus lervia*) experimentally infected with *Mycoplasma ovipneumoniae* (Movi group, A-C) or nasal lavage fluid from *M. ovipneumoniae* PCR positive domestic sheep (Wash group, D-F). (A) AD #4. Multiple fibrous adhesions spanning from the costal parietal pleura to the visceral pleura of the right cranioventral lung lobes. (B) AD #1. The ventral portions of the cranial aspect of the right cranial and the right middle lung lobes are dark red and consolidated with a single pleural fibrous adhesion on the cranial aspect of the right cranial lobe. (C) AD #1. An accumulation of mucopurulent exudate is within the tracheal lumen. (D) AD #14. Approximately 70% of the right lung lobes and corresponding costal parietal pleura are covered by a severe, fibrinous pleuritis with serosanguinous pleural effusion. The lungs are diffusely dark red. (E) AD #14. On cut surface, the lung parenchyma contains multiple to coalescent, slightly firm, irregular marbling corresponding to fibrinosuppurative and necrotizing bronchopneumonia. Note the exudate within a bronchus (arrowhead) and the thick layer of fibrin covering the pleura (arrow). (F) AD #13. Multifocal abscesses replace the pulmonary parenchyma. The remaining parenchyma is dark red, and the airways are filled with purulent exudate.

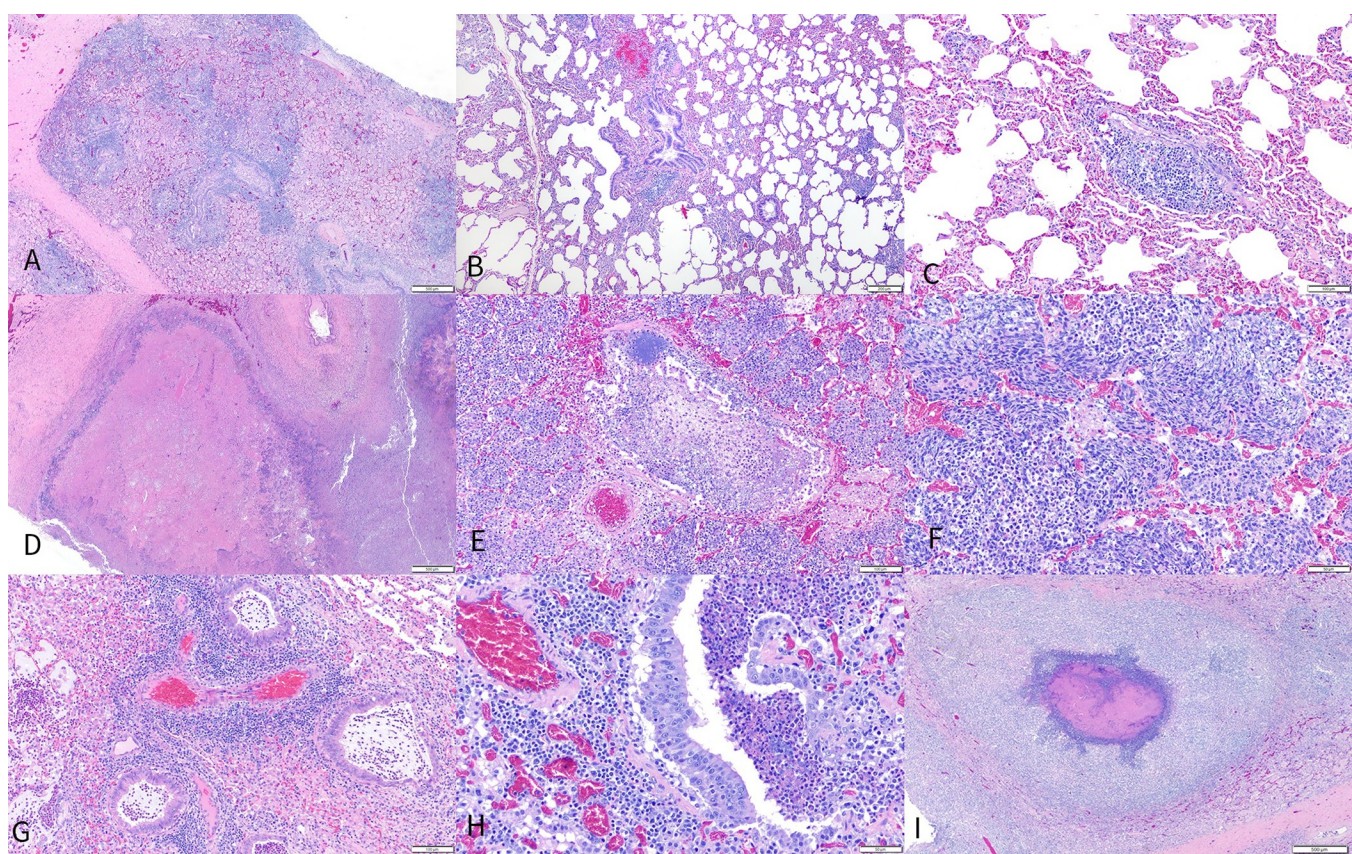

**Fig 3. Histologic pulmonary patterns observed in aoudad (*Ammotragus lervia*) experimentally infected with *Mycoplasma ovipneumoniae* (Movi group) or nasal lavage fluid from *M. ovipneumoniae* PCR positive domestic sheep (Wash group).** (A-C). Pattern 1: Chronic lymphohistiocytic, bronchointerstitial pneumonia with BALT hyperplasia and interlobular fibrosis (A), alveolar septal expansion by lymphocytes and macrophages (B), and perivascular lymphoid accumulations (C). (D-F). Pattern 2; Acute fibrinosuppurative bronchopneumonia with large areas of coagulative necrosis surrounded by a rim of degenerate neutrophils (D), intralesional bacterial colonies (E), and oat cells (F). (G-I). Pattern 3: Superimposition of patterns 1 and 2 (G) with bronchiolar epithelial hyperplasia (H) and abscess formation (I).

AD nasal swabs. Specifically, we were unable to use MLST to characterize the *M. ovipneumoniae* strain(s) present in either the Movi or Wash group inoculum. While these challenges prohibit formal confirmation of transmission of any specific *M. ovipneumoniae* strain between groups or individuals, the combination of serologic and molecular patterns presented strongly suggest AD can harbor, shed, and intraspecifically transmit *M. ovipneumoniae*.

Lung samples from 3/3 Control-contact AD (AD #15, 16, 21), 4/6 Movi AD (AD #1,3,5,7), and 1/5 Wash AD (AD #13) tested *M. ovipneumoniae* qPCR negative, but all 8 negative individuals had lesions consistent with *M. ovipneumoniae*, 7 were seropositive, and 6 gave positive qPCR nasal swab results 0 to 7 days pre-euthanasia. This inconsistency between qPCR results shortly before and after euthanasia may suggest that AD cleared the infection, or that DNA extraction quality/quantity differed between swab and tissue samples. This may be an important consideration for future studies investigating respiratory agents in AD that utilize pre and/or post-mortem samples. More work is needed to determine if, like BHS, interindividual variation exists in AD pathogen carriage propensity [43]. This will be crucial for determining whether transmission dynamics, particularly the potential for superspreading events/ individuals, differ between AD and BHS [43].

Notably, this is the first investigation utilizing cELISA to describe *M. ovipneumoniae* exposure patterns in AD, and much work is needed to describe the utility and limitations of this

assay in this species. Intermediate longitudinal samples were not collected from control-contact AD after Movi and Wash group inoculations to minimize capture related morbidity/mortality, but 67% (2/3) of Control-contact AD were *M. ovipneumoniae* seropositive upon study termination, demonstrating intraspecific transmission of immunogenic doses of *M. ovipneumoniae* between AD. The single seronegative control-contact AD (AD #15) tested qPCR indeterminate upon study termination (day 86), indicating that seroconversion may have been ongoing in this individual, or that its titer was low given an apparently sub-immunogenic amount of *M. ovipneumoniae*. Control-contact AD were often seen within 5 meters of the fence-line shared with Movi group AD while Wash group associations were infrequent. Considering *M. ovipneumoniae* can be transmitted via aerosol droplets at least 20 meters [44, 45], it is surprising that transmission and subsequent seroconversion of control-contact AD was incomplete (2/3; 67%) upon euthanasia (86 days post-arrival). Alternatively, use of a BHS strain may have reduced the virulence, transmissibility, and carriage propensity in *Caprinae* compared to DS strains [40], but we were unable to determine which strain(s) control-contact AD were exposed to due to poor MLST amplification and sequence non-convergence. The development of chronic lesions and the detection of *M. ovipneumoniae*-specific antibodies in control-contact AD suggests these individuals were likely exposed at least 14 days prior to sampling (approximately 34–48 days p.i for the inoculated AD groups) and developed mild lesions with subclinical disease. Overall, more work is needed to experimentally and definitively confirm that *M. ovipneumoniae* can be transmitted among AD.

One Movi (AD #2) and one Wash group (AD #9) AD were *M. ovipneumoniae* seropositive upon capture, but due to the high insufficient sample rate with 16S sequencing used at capture (50% insufficient samples), we were unable to determine if either of these AD were natural carriers before the study. Subsequently, we could not discern the roles of potential natural *M. ovipneumoniae* transmission versus experimental inoculation within our study. We handled and/or collected each group's inoculum prior to collecting samples on day 0 (inoculation day) for each group. This logistic and personnel constraint introduced a contamination risk for day 0 samples, but MLST results did not permit discrimination between potential natural strains and inoculum strains. Regardless of pathogen source(s), our results still indicate AD are competent *M. ovipneumoniae* hosts for extended periods of time (at least 53 days p.i) even after seroconversion. While validated antibody persistence data and diagnostic criteria for AD are currently lacking, the efficient seroconversion seen in this study suggests that free-ranging AD can be monitored for exposure to *M. ovipneumoniae* using common serological surveillance techniques for BHS and DS. *M. ovipneumoniae* carriage and shedding capacity are strain specific in many *Caprinae* [24, 25]. Under current knowledge, the presence of different *M. ovipneumoniae* strains between treatment groups could cause significant differences in transmissibility, clinical severity, and pathology in *Caprinae* members [24, 25]. Additionally, we were unable to quantify the *M. ovipneumoniae* dose present in each inoculum, and it is unknown how this added variability impacted our results.

AD can harbor *Pasteurellaceae* species important in BHS pneumonia etiology regardless of whether they were experimentally inoculated with them. However, the high rate (63%) of mixed culture results pre-inoculation potentially biased our estimation of pre-inoculation *Pasteurellaceae* presence and composition via culture, isolation, and species identification by MALDI. *Biberstinia trehalosi* and *Mannheimia haemolytica* are both potentially leukotoxigenic, but the latter is thought to carry the leukotoxin-A gene more frequently [25]. *M. haemolytica* was more common in the Wash group, likely due to the high carriage rate of this species by DS [30]. While only lung samples were tested for the leukotoxin-A gene, its exclusive detection in the Wash group suggests the DS inoculum was a reasonable analog to previous DS studies of BHS pneumonia etiology [23, 25, 36]. The increasing p.i detection frequency of *B.*

*trehalosi* in the Movi group and *M. haemolytica* in the Wash group throughout the study suggests that *Pasteurellaceae* bacteria colonized, expanded, or at least persisted in the respiratory tract, presumably due to disruption of the mucociliary apparatus by *M. ovipneumoniae* [46, 47]. The increasing p.i detection of *T. pyogenes* in tonsil swabs may also reflect the pattern of opportunistic infections afforded by *M. ovipneumoniae* colonization. Our results demonstrate that AD are susceptible to respiratory disease after exposure to leukotoxigenic *Pasteurellaceae* and that coinfection with *M. ovipneumoniae* produces disease more severe than with *M. ovipneumoniae* alone. We did not assess the full spectrum of microbiome changes that may have also contributed to pathogenesis. However, the objectives of this study were focused on pathogens with known involvement in BHS pneumonia epizootics. The potential involvement of other agents in the development of observed lesions was beyond the scope of this study.

As expected, inoculated AD exhibited significant changes to leukocytes, platelets, and acute phase proteins typically associated with innate and adaptive immune responses to infectious diseases [48, 49]. The Wash group inoculum apparently caused more severe leukocyte and acute phase responses compared to the Movi group [50], suggesting that DS-sourced *Pasteurellaceae* (*M. haemolytica*) affects pneumonia pathogenesis in AD just as in BHS [23]. Specifically, it is known that *M. ovipneumoniae*–*M. haemolytica* coinfection induces a strong, synergistic neutrophilic response [51]. Patterns in CMS corroborate this: Wash group AD exhibited earlier and more severe clinical signs throughout portions of the study. Elevated total leukocytes, neutrophils, platelets, and plasma protein levels early in pathogenesis preceded worsening clinical disease as expected. Concurrently, elevated lymphocytes signaled apparent clinical improvement.

All AD (n = 14) exhibited clinical signs and lesions consistent with mild to severe respiratory disease. While acclimatization stress likely exacerbated pathogenesis [52, 53], disease severity varied greatly. As expected, only the Wash group had lesion patterns consistent with leukotoxigenic *Pasteurellaceae* infection characterized by the formation of oat cells. Lesions suggestive of *M. ovipneumoniae* infection [54, 55] (pattern 1) were present in all control-contact AD despite detecting the agent in only one control-contact AD. Further, serum samples indicate that AD #15 did not develop a measurable antibody response to *M. ovipneumoniae* despite suggestive lesions and close contacts with naturally exposed pen-mates. These inconsistencies suggest that future AD investigations should further characterize the shedding dynamics and immunologic responses of AD to *M. ovipneumoniae* and *Pasteurellaceae* over a longer period using repeated sampling of exposed individuals. Overall, the Wash group exhibited more rapid and severe clinical signs and more severe pneumonia compared to Movi group AD. While we did not perform virus isolation or PCR to detect active viral infections, serologic evidence of BRSV exposure prevents us from excluding its participation in observed results, although no lesions suggestive of active BRSV infection were observed.

This study demonstrated that AD can harbor and transmit *M. ovipneumoniae* intraspecifically, which may present a risk to BHS populations they share the landscape with in the Trans-Pecos region of Texas. Much work is needed to determine the disease risk that free-ranging AD pose to BHS. Aoudad experienced pneumonia after exposure to *M. ovipneumoniae* plus native AD lung flora or *M. ovipneumoniae* plus DS-origin leukotoxigenic *Pasteurellaceae*. However, the spectrum of potential agents was not assessed in this study, as the primary objective was to assess host competency for *M. ovipneumoniae* and leukotoxigenic *Pasteurellaceae*. The suspected role of acclimatization stress in our study combined with a lack of pneumonia reported in *M. ovipneumoniae*-exposed AD in the United States suggests clinical disease may not be as severe in free-ranging individuals. Future work should focus on determining the pathogenicity and virulence of *M. ovipneumoniae* and leukotoxigenic *Pasteurellaceae* transmitted from AD to BHS as well as epidemiological patterns of these pathogens in free-ranging

AD. Because virulent DS-origin pathogens were used in this study, future efforts should specifically focus on describing the *M. ovipneumoniae* and *Pasteurellaceae* strain compositions of free-ranging AD. This will be crucial for determining both the transmission dynamics and the demographic implications of potential spillover from AD to BHS. Further, investigating strain-specific sources and virulence (potential spillover from domestic *Caprinae* vs. natural carriage) of *M. ovipneumoniae* in naturally exposed free-ranging AD will help determine what population and community level effects wildlife managers may reasonably expect in the southwestern U.S and the Trans-Pecos region of Texas in particular.

## Supporting information

**S1 Fig. Longitudinal blood parameters of inoculated and control aoudad.** Longitudinal assessment of: A, median white blood cells (K/uL); B, lymphocytes (K/uL); C, neutrophils (K/uL); D, monocytes (K/uL); E, platelets (K/uL); F, total plasma proteins (g/dL); and G, fibrinogen (mg/dL). The group level median of each value was assessed for each day P.I (days postinoculation. Movi group aoudad were inoculated with only *Mycoplasma ovipneumoniae*; Wash group aoudad were inoculated with *Mycoplasma ovipneumoniae*-containing domestic sheep nasal washes; control-contact aoudad were not inoculated, but were allowed contact with the other groups.
(TIF)

**S1 Appendix. Multi-Locus Sanger sequencing Appendix.** Multi-locus sequencing results for the inoculum and aoud swabs. This includes descriptions of failed amplification and nonconverged sequences.
(DOCX)

**S2 Appendix. Pathology Appendix.** Supplementary details on gross pulmonary and non-pulmonary lesions of inoculated and control aoudad.
(DOCX)

**S1 Table. Individual lesion patterns.** Supplementary details on the pattern and distribution of lesions and pathogen detections for individual aoudad.
(DOCX)

**S1 Dataset. Blood parameters.** Data file of blood parameters collected.
(XLSX)

**S2 Dataset. Daily clinical scores.** Data file of clinical scores collected.
(XLSX)

## Acknowledgments

We first express our sincere appreciation to Houston Safari Club and the Texas Bighorn Society for logistically supporting this work. A great debt of gratitude is owed to Texas Parks and Wildlife Department's Big Game Program staff, who coordinated the capture of our study animals. Destiny Taylor and all Texas A&M University's Veterinary Medical Research Park staff were instrumental in pen construction and coordinating animal housing arrangements. We thank the staff, Alen Merdzo and Joshua Freeman for their technical support with necropsy and tissue disposal. Finally, I sincerely thank all my volunteers: Jamie Benn, Jeremy Echols, Cora Garcia, Jordan Gomez, Emma Haschke, Kelly Head, Aaron Nunez, Brianna Stofas, and Sydney Tejml. The late Bob Dittmar of Texas Parks and Wildlife was instrumental in planning and coordinating several research projects aimed at addressing wildlife health across the state.

His generosity, kindness, and influence on conservation will never fade on us. Not only was he instrumental in this work, but he has had a profound influence on these authors, our perspectives on natural resource protection, and bettered everyone who had the pleasure of working with him. We take it upon ourselves to honor his legacy by continuing research in wildlife health and constantly offering our experiences to teach the next generation of wildlife managers.

## Author Contributions

**Conceptualization:** Logan F. Thomas, Robert O. Dittmar, Froylán Hernandez, Walter E. Cook.

**Data curation:** Logan F. Thomas, Walter E. Cook.

**Formal analysis:** Logan F. Thomas, Raquel R. Rech, Walter E. Cook.

**Funding acquisition:** Logan F. Thomas, Robert O. Dittmar, Froylán Hernandez, Walter E. Cook.

**Investigation:** Logan F. Thomas, Chase M. Nunez, Raquel R. Rech, Walter E. Cook.

**Methodology:** Logan F. Thomas, Robert O. Dittmar, Walter E. Cook.

**Project administration:** Logan F. Thomas, Robert O. Dittmar, Walter E. Cook.

**Resources:** Logan F. Thomas, Walter E. Cook.

**Supervision:** Logan F. Thomas, Robert O. Dittmar, Froylán Hernandez, Walter E. Cook.

**Validation:** Logan F. Thomas, Dallas Clontz, Raquel R. Rech.

**Visualization:** Logan F. Thomas, Dallas Clontz, Raquel R. Rech.

**Writing – original draft:** Logan F. Thomas, Dallas Clontz, Chase M. Nunez, Raquel R. Rech, Walter E. Cook.

**Writing – review & editing:** Logan F. Thomas, Dallas Clontz, Chase M. Nunez, Raquel R. Rech, Walter E. Cook.

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
