## [Decision Letter · Decision Letter 0]

15 Aug 2023

PONE-D-23-16154Evaluating the transmission dynamics and susceptibility of aoudad (Ammotragus lervia) experimentally infected with Mycoplasma ovipneumoniae and leukotoxigenic PasteurellaceaePLOS ONE

Dear Dr. Thomas,

Thank you for submitting your manuscript to PLOS ONE. After careful consideration, we feel that it has merit but does not fully meet PLOS ONE’s publication criteria as it currently stands. Therefore, we invite you to submit a revised version of the manuscript that addresses the points raised during the review process.

The three reviewers have recommended major revisions. The missed many details of your experimental procedure that will be necessary to be provided for considering your work for publication. On the other hand, I think it would be worthy to include a little paragraph about the potential impact of Aoudad on native ungulate populations as is happening in southeast Spain (Cassinello, et al. 2004. Range expansion of an exotic ungulate (*Ammotragus lervia*) in southern Spain: ecological and conservation concerns. *Biodiversity and Conservation*
**13**, 851–866) with aoudad populations acting as reservoirs for sarcoptic mange (González-Candela et al. Population effects of sarcoptic mange in Barbary sheep from Sierra Espuña Regional Park, Spain. J. Wild. Manage. 40(3): 456-465).

Regarding the blood analysis, it keeps unclear to me whether you got blood samples for haematology at each sampling point or just at beginning and end of the study. Please, explain this. If you have got blood at each sampling point, I would recommend analysing the dynamic of your blood parameters. In any case, there is very little information about this haematological section neither in the methods nor in the results section.

Please, provide references for tooth eruption patterns. Ungulates are generally classified as Kids, juveniles, yearlings sub-adults (also prime-age) and adults.

We look forward to receiving your revised manuscript.

Kind regards,

Emmanuel Serrano, PhD

Academic Editor

PLOS ONE

“We first express our sincere appreciation to Houston Safari Club, Texas Bighorn Society, and The Wild Sheep Foundation for funding this work. “

“This study was funded by the Wild Sheep Foundation's Grant-in-Aid Program (L.F Thomas; W.E Cook).  The funder had no role in study design, data collection and analysis, decision to publish, or preparation of the manuscript.”

Reviewers' comments:

Reviewer's Responses to Questions

**Comments to the Author**

1. Is the manuscript technically sound, and do the data support the conclusions?

Reviewer #1: Partly

Reviewer #2: No

Reviewer #3: Partly

2. Has the statistical analysis been performed appropriately and rigorously? 

Reviewer #1: No

Reviewer #2: No

Reviewer #3: Yes

3. Have the authors made all data underlying the findings in their manuscript fully available?

Reviewer #1: No

Reviewer #2: Yes

Reviewer #3: No

4. Is the manuscript presented in an intelligible fashion and written in standard English?

Reviewer #1: Yes

Reviewer #2: Yes

Reviewer #3: Yes

5. Review Comments to the Author

Reviewer #1: Evaluating the transmission dynamics and susceptibility of aoudad experimentally infected with Mycoplasma ovipneumoniae and leukotoxigenic Pasteurellaceae

Logan Thomas and others

Summary and general comments: Seeking to determine the susceptibility of aoudad to Mycoplasma ovipneumoniae alone or in combination with Pasteurellaceae, and their susceptibility to respiratory disease caused by these agents, the authors captured wild aoudad, separated them into three pens to house 6 aoudad challenged with an M. ovipneumoniae cultured from infected bighorn sheep, 5 aoudad challenged with nasal washings from domestic sheep previously found to shed M. ovipneumoniae, and 3 control (unchallenged) aoudad with fenceline contact to the two groups. Seven other aoudad intended for the study and captured at around the same time died or were euthanized during the 38d acclimatization period after capture before these studies were started. The wild aoudad likely were already exposed to M. ovipneumoniae, as the diagnostic testing done at capture used a metagenomic method of unreported sensitivity, which was unable to be completed on half of the captured aoudad. Furthermore, M. ovipneumoniae was detected in most of the aoudad just pre-challenge. Seroconversion prior to challenge was also observed, although this is difficult to interpret given the unknown sensitivity and specificity of the cELISA test in this unvalidated host species. M. ovipneumoniae was detected in the challenged sheep through the duration of the study. These findings confirm that aoudad are competent hosts of M. ovipneumoniae, but because infection was documented prior to the study challenge and because no strain typing data is presented, the authors’ claim that transmission occurred among the aoudad in this study, although not unlikely, is not directly supported.

I am a little concerned that the control group was excluded from the clinical signs comparison. The authors indicated that no clinical signs were observed in the control group, but Table S3 does identify at least some clinical signs in two individuals in this group.

The use of MLST to identify strain types is described briefly in the M&M but no data are presented in the Results. The discussion indicates that ‘non-convergence’ resulted in the inability of this test to be used. I cannot find any of the MLST data in the supplemental tables either.

The paper would benefit greatly by editing for brevity, clarity, and readability.

Materials and methods:

- Kindly provide more information about the 7 animals that died or were euthanized prior to the study, and these may provide insight into the disease and respiratory infections of the wild herd. Were they sampled for microbiologic testing? Were lung lesions observed? What was the trigger for the 3 euthanasias?

- Kindly provide more information about the challenges. In line 139-144 describing the origin of the challenge inoculum, no data is provide to confirm the growth of M. ovipneumoniae from the source anmals. The dose and method of challenge inoculation of the first group is not described. In line 144-147, were the domestic sheep sources of the nasal washes re-tested at the time of collection to confirm the continued presence of M. ovipneumoniae? Were the challenge washes from individual domestic sheep or were they pooled prior to challenging the aoudad?

- Regarding the microbiologic testing during the project: Has the cELISA assay performance been validated in aoudad? I know it has been found to not work at all well in another Caprinae species, domestic goats. Can you clarify why it was only conducted on samples beginning 14 or 21 d post-challenge? Were no blood samples available from the time of capture or challenge? Can you explain why the attempt at MLST was done on the ‘last nasal swab’ sample, rather than on the DNA extract the lab had obtained from the earlier PCR detection?

- Regarding the 16S metagenome data acquired from the aoudad samples taken at the time of initial capture, were you able to access the raw sequence data from the company that performed the testing, so as to query the dataset directly for M. ovipneumoniae 16S sequences? If not, can you confirm their data analysis methods would have detected these veterinary pathogens, if present?

- L179-180 Were the lung specimens submitted to WADDL selected to contain medium – large airways, the site of M ovipneumoniae infection?

Results:

L206 – 246 This data presentation is very difficult to follow. In general, I suggest using Table 1 to show all detailed sub-group specific results, and use the text to simply identify group- or timepoints which differed significantly. The data in the table should be complete – that is, the data from timepoint 0 d should be added.

L214-217. If I am understanding this correctly, the reader has to compare the study wide tallies with the post inoculation tallies to deduce that Movi was detected in at least 4 aoudad in the Movi group prior to the day of inoculation. Is that correct?

L218-219 Please add this data to Table 1. I also think it would help readability to move this sentence to be the first in the Results section.

L219-221 Please explain a little more. Was the ‘re-analysis’ done by you or by the laboratory that ran the tests? Is anything known about what caused this discrepancy?

L229 Table 1: Please add Day 0 results to Table 1. Indicate the lack of rtPCR data for the capture time point, and the method actually used at this time. For the Capture column, using the total number of animals as the denominator rather than the numbers for which results were obtained would be more clear. For example, the Movi group would say 0/2 (5 Ins). My thinking is that if insufficient samples were obtained from some animals, negative results should not be tabulated.

L236-250 Like the results for the ‘Novi’ groups, I suggest that the details for the ‘Wash’ and ‘Control’ groups be left in the table, and only the significant effects detected be presented in the text.

L242 Should still cite Table 1, not Table 2.

L252-261 Like the PCR results, I suggest leaving the details to Table 2, and use the text to only report the summary results (i.e., the final two sentences).

L274-313: Like above, please report the data in the table, and save the text for summary findings and significant comparisons.

L315-342: Same comment as previous.

L347-350: Why all the differing euthanasia dates unbalanced across treatment groups?

L353: define CMS. Also, please explain what data are added to make the scores ‘cumulative’.

L367-431: Suggest deleting most of the detailed textual description. Perhaps just present Figure 3 then retain the final paragraph summarizing the patterns by experimental group. (If considered important, perhaps the details could be moved to an appendix?)

Discussion

The Discussion is very long, and quite speculative in places (further noted below). I think it would improve this paper to shorten it, tighten the targets of discussion to areas where this study moves the field forward, and edit it to improve clarity and readability.

L433-436 Suggest adding the observation of pre-existing infection in aoudad at capture, indicating infection by natural processes.

L436: The data presented may be consistent with intraspecific transmission, but they are not confirmatory: it is difficult or impossible to rule out persistent infections fluctuating above and below the detection threshold for PCR. However, if the inoculum strain types of M. ovipneumoniae, obtained pre-challenge, had been detected in experimental groups that had not received those challenges, that would have been direct evidence of transmission.

L437-440: Unless lung specimens had been selected for presence of large ciliated airways, the negative results from some animals do not rule out lower respiratory M. ovipneumoniae infection.

L448: This seroconversion data may be the first evaluation of the cELISA test in the aoudad species. Worth mentioning?

L461: There is no multi-locus sequence data presented in the Results. Also, define ‘non-convergence’.

L464: “…to confirm that M. ovipneumoniae is transmitted among aoudad” as I believe the data in this study is suggestive but not confirmatory.

L468: What possible sources of M. ovipneumoniae infections detected immediately pre-challenge exist, other than naturally occurring infections?

L470-471: Multi-locus sequence typing would most likely be able to discriminate between these sources, but no data is presented in Results.

L478-491: This paragraph is largely speculative and I suggest deleting it, to shorten a very long discussion.

L499-501: I could not find any lktA detection data in the Results.

L506-9 I could not find any data showing that the Wash group was exposed to leukotoxigenic Pasteurellaceae in the challenges.

L526-528: The lab (WADDL) that runs this test cautions that the test is validated at the group level, not in individuals, because individual non-reactors have been observed in all validated subject species.

Reviewer #2: The authors ran an experiment to see if Auodad can become infected with M. Ovi and Pasteurella, pathogens relevant to bighorn sheep. They found Auodad can become infected and do develop lesions.

There are several oddities in the experimental design that the authors should clarify the reasons for (i expect practical). Why the different periods of time for acclimitization between the groups ? The groups sizes are quite different - the transmission group would be very important as the authors are trying to make the point that auodad are a risk to bighorn sheep but it was only 3 individuals -why? Why was ivermectin given on day 45 and not upon initial capture? How many of the animals had parasites upon deworming, and could this have affected the results (knowing that micro and macro parasites interact through the immune system)?

My main concern is that the title and abstract discuss the potential for transmission, however for M Ovi the only diagnostic reported is PCR, which includes live and dead bacterial signal. How do they know if the animals are able to transmit live pathogen? i.e. how can they assess risk to sympatric bighorn. The authors showed conclusively that auodad can get infected but I would argue the evidence for transmission is quite weak (1/3 sheep became PCR positive in the contact group)? Or are they arguing that because they took nasal washes from infected sheep and then moved those into susceptible sheep that that is a proxy for transmission potential? I assume the dose received in a nasal wash is much higher than in a normal sheep to sheep interaction? Some careful thought and discussion should be given to this potential. A mathematical model of transmission potential, using this experimental data as estimates, could help strengthen the transmission statements if the authors wish to.

Reviewer #3: The article PONE-D-23-16154 is a study focused on bronchopneumonia in the Aoudad (Ammotragus lervia), which is an invasive species in north America that shares its range with the native Bighorn sheep (Ovis canadensis) in Texas. Bronchopneumonia is a population-limiting disease in Bighorn sheep in which M. ovipneumoniae has a primary etiological role, but other bacteria from the Pasteurellaceae family are also involved.

The authors did an experimental infection of wild captured Aoudad to assess if this species is susceptible to develop disease from domestic sheep nasal washing and from broth-cultured nasal swab samples from bighorn sheep. The topic is of interest for understanding better the pathogeny of this polymicrobial disease in another wild ungulate, however, in my opinion the study has a poor experimental design with major flaws that the authors need to clarify or address before publication. One of the main concerns is that the study failed to demonstrate if the animals incorporated in the study were free of M. ovipneumoinae before the experimental challenge. In the methods they mention they used a next-generation sequencing method, without further specifications, to discard the presence of M. ovipneumoinae in nasal swabs. However, the method failed in about half of the animals. Did the authors tried to different dilutions to avoid inhibitors? Did you try a specific rt-PCR for Movi as used in subsequent samples? This should be a more sensitive method as typically amplifies shorter DNA fragments, and it seems you did not have inhibitions in subsequent sampling. Why you do not show the results of the 16SrRNA sequencing to describe the nasal microbiome prior the challenge? The status of the animals before the challenge is critical and it is not well understood in the article, the authors should confirm either by serology (at least) or molecular detection the free status of all animals and clearly state it in the manuscript. Currently it is not known if the animals that with invalid results in nasal swabs were at least seronegative. Another concern is that, according to the methods described, the authors did not test what was present in the inoculums used (nasal wash and broth culture from a nasal swab), neither the dose of M. ovipneumonniae, the presence of other bacteria in the culture-broth inoculum or the lkt-A virulence gene as the title of the article claims. The authors need to either clarify these issues or preferably, perform further analyses on the stored inoculums. The methods are clumsy and poorly described, I had to read back and forth some sections and I am still not sure about some of the methods used because they do not fully correspond with the results presented or are poorly or not described at all. Both the title and the methods mention the detection of lkt-A virulence gene, but I have not found these results in the manuscript.

The authors need to rewrite and do a better effort to clearly present the study design because the manuscript, as it is written, is almost impossible to understand. Some other major flaws are impossible to amend, for example, the control-contact group was not sampled until the end of the study period, or that an antiparasitic treatment was applied in the middle of the experimental infection and it is unknown how differences on parasitism could have affected the response of the challenges. This should however be properly discussed.

Specific comments:

- Line 46-47: what is an exotic or native interaction?

- Line 47-49: I am not a native English speaker, but verb tenses does not seem to match in this sentence.

- Line 58-61: please add a reference to the statement.

- Line 78-80: please clarify if you refer to bighorn sheep in this sentence.

- Line 80: this is based on targeted studies? Passive/scanning surveillance?

- Line 80: prior to what?

- Line 83-84: in my opinion you did not test the susceptibility to M. ovipneumoniae with or without co-exposure to Pasteurellaceae, because you used an inoculum with unknown infectious agents and doses. Please review your objectives in relation to the study design and methods.

- Line 86: transmission or shedding?

- Line 101: please define the reasons for euthanasia.

- Line 107: “control-contact” may define better this experimental group, as this is not exactly a control group.

- Line 112: acclimatation period? But inoculations were already done?

- Line 114: please explain the methods to estimate the body condition daily. Were these methods the same as the ones used on capture events?

- Line 121: please correct to “captures”. I think “initial” does not add any significant information or clarification, but it raises the question if the capture methods in Carrizo mountains have changed over the study.

- Line 124-125: what do you mean by thereafter? Every 30-45 minutes? Please review the sentence, it is not clear. If there were consecutives doses and a defined period for acclimatation to captivity, please specify in the sentence.

- Line 126-127: The way it is written is confusing. Instead of linking the methods with a period of time, which has no logical connection (“after acclimatation”), please mention the purpose of those chemical immobilizations. At this point, the reader does not know why you are doing these immobilizations.

- Line 127: How was stress handled during those immobilizations? And how was chemical immobilization applied? Were the animals physically restrained before, on a crate, with guns, etc… please provide more details on the methods specially focusing on how stress was managed.

- Line 134: please clarify if ivermectin was applied to all experimental groups. Why was the treatment not applied during the acclimatation period and before the infection?

- Line 141: I believe the modified tryptic soy broth is not a specific medium for the growth of mycoplasmas. Why did you choose this medium? How did you discarded the growth of other bacteria than M. ovipneumoniae.

- Line 139-144: How the inoculum was quantified for M. ovipneumoniae? Was the culture broth tested for other bacteria? Where other mycoplasmas discarded? Please clarify.

- Line 145: please make sure that you have explained what is “DS”. I am not sure if this was mentioned before.

- Line 147: qPCR and rt-PCR are abbreviations to the same method, either referring to the concept of “quantitative” or “real-time”. You previously used the standardized abbreviation of qPCR, if you also refer here to the same method please use the same abbreviation. It is weird to use different abbreviations for the same method, even if the primers and probes used are different.

- Line 149: why the Wash inoculum was also administrated into conjunctival sacs? Is this a demonstrated infection route for M. ovipneumoniae? Where the authors also interested on demonstrating the pathogenic effect of other bacteria on ocular and conjunctival tissues? Please explain.

- Line 150-151: Negative to what sample? It is not explained and even if it is explained later in the text it should be clarified in this sentence. What is the rationale for performing the re-inoculations? Even if the animals presented respiratory clinical signs were re-inoculated?

- Line 160-161: why antibodies against M. ovipneumoniae were not tested at day 0 and consecutive days to test for serconversion? This information must be included as it can affect the results of the experimental infection.

- Line 163: from what sampling?

- Line 163-167: The Muller-Hintor medium is non-specific por Pasteurellaceae, how were the colonies selected for further identification? How were the isolates identified? Please describe properly the methods used.

- Line 167: please explain properly the experimental design, including sampling days for each group, pathogen, etc…

- Line 170-171: please provide the reference for the PCR used and the threshold used for determining the sample’s Ct.

- Line 171-172: MLST is usually performed from pure cultures, not field samples. If there is more than one strain the approach is useless. Please explain better the methods.

- Line 173: please describe the details of the next-generation sequencing, this is an umbrella term for different methods. Was the 16SrRNA gene full-length sequenced? Overall, it is surprising that the authors used a next generation sequencing just to discard the presence of M. ovipineumoniae in nasal swabs. Why the authors did not use the results on the detection of other bacteria that could be involved in respiratory disease (Pasteurellaceae, Moraxella spp., etc.)?

- Line 179: what sections of lungs? From all the lung lobes? Only from lesions?

- Line 182: please specify which tissues were cultured.

- Line 187-189: this is not clear. What time periods? Do you mean sampling days?

- Line 189: What groups were compared? Sex? Age? You need to be always specific. If the you compared the “Movi”, “Wash” and Control group you need to mention it.

- Line 191: how many times was the clinical scores recorded per individual and day to calculate the mean?

- Line 194: How were compared?

- Line 202-203: this sentence is not necessary and weird given that the authors did not mention anything about degrees of freedom before.

- Line 206: three fence-line is not specific to the distance of separation. Please add this .information.

- Line 208: please specify to what the samples were possible.

- Line 210-211: by what method? If it was by metagenomics?

- Line 212: I understood there were two treatment groups and a contact control?

- Line 213: positive to what? You need to be specific in each sentence to avoid any doubt. Please correct.

- Line 214: what is the different between swabs collected post-infection and study-wide? This is confusing. I would assume that study wise also includes the samples collected at field captures.

- Line 206-217: this paragraph is impossible to follow, please rewrite to be clearer with the methods.

- Line 218: does this mean that the “Movi” group already had Mycoplamsa ovipneumoniae before inoculation? You mention you had 7 positives but I believe there were 6 animals in that experimental group? Please clarify.

- Line 222: seven days post-infection again? You need to use a clearer explanation of the methods.

- Line 223: what analyses? And why you mention it here?

- Line 224: please make sure you correctly explain the experimental design in the methods, including the sampling days per experimental group etc.

- Line 229: in the results section you have 14 animals divided in three experimental groups, but in the table there are 15 animals at capture. I assume you had casualties before the inoculation to explain this mismatch, but in this case, I suggest removing them from the table to keep consistency on the number of animals you incorporated in the experimental study. The results of animals not incorporated in the experimental infection you can mention them separately in the text.

- Line 229: In the methods of the article the detection of M. ovipneumoniae at capture is perfromed by amplifying the 16SrRNA using a next-generation sequencing, but here you mention that you detected it by rtPCR. Please clarify, or correct.

- Line 230: I don’t think percentage is appropriate for such low numbers. I suggest to include the proportions as the main result, and the authors can decide whether they still want to keep the percentage in brackets.

- Line 230: the format of the table in the pdf I have is awkward and not the same in all columns and rows. Perhaps this is an artifact in the generation of the pdf, but please make sure you comply with the journal guidelines for tables.

- Line 231: In my opinion, this should be included in title of the table, not as a footnote. Please describe “P.I.” and “NA” in the footnotes.

- Line 236: positive to what? What is insufficient? Is this information any relevant? Why does not match with the total P.I. column in table 1? Even adding the swabs at captures does not sum up to 30.

- Line 239-240: you had positive samples to M. ovipneumoniae before the inoculation? Please clarify?

- Line 240-243: this information is redundant with the table, please remove and reference to Table 1.

- Line 242: if I understood well this information refers to Table 1, not Table 2.

- Line 243-244: in line 236 you state that throughout the study you detected 50% of positive samples in the Wash group. What is the difference between “throughout” and “overall”? You should use clear terms and consistent throughout the article.

- Line 245: “after the Movi and Wash group were inoculated” is not a precise point of time, please be specific.

- Line 250-251: In my opinion, this test statistic is not necessary.

- Line 267: Please see my comments on Table 1 because many still apply in this table.

- Line 280: why mixed infections could not be identified? I understood that you did microbiological cultures for obtaining pure cultures. Furthermore, I believe I have not read the methods on how you identified or detected Pasteurellaceae. Please review all methods and make sure that are all clear.

- Line 281: how swab samples preinoculation sum up to 27 if you have 15 individuals divided in three experimental groups? I can not understand either by looking at the numbers of the table. Please explain or make it clearer in the text.

- Line 285: please review my previous comments on Table 1 because some also apply to Table 3.

- Line 293-298: Please remove redundant information that already appears in table 3.

- Line 304: please change “detection” by “isolation”.

- Line 305-307: what experimental groups?

- Line 310: more frequently than what other group?

- Line 313: in the materials and methods you mention that you did a PCR for the presence of the lkt-A gene, but you do not provide these results. Can you clarify this?

- Line 311-313: the culture methods used are not very sensitive for the detection of diverse Pasteurellaceae, especially considering culture inhibitions among them (see Kugadas et al., 2014.Veterinary Microbiology, 174:1-2, pp 155-162.). In my opinion, it has not much sense to compare specific differences on the isolation of different Pasteurellaceae for specific sampling days.

- Line 343: I suggest moving this section to the beginning of the results because it can help to understand why the animals sampled throughout the study vary.

- Line 344: you also inoculated the conjunctival sac; no ocular lesions were detected over the course of the experimental infection?

- Line 348 and line 349: please explain the reasons why the animals 1 and 13 were euthanized.

- Line 360: why the CMS drop suddenly in the Wash group at around 38 dpi? Please explain briefly in the table footnote or title, because this trend is not possible to understand without explanations.

- Line 379: what about pathological lesions in the lungs of the control-contact group?

- Line 398: did you perform cultures from the caseous lymphadenitis or other lesions observed a part of the lungs?

- Line 436: the transmission between aoudad groups probably happened before 53 d.p.i, as the animals had seroconverted, therefore “for at least” does not sound completely correct here. Please review.

- Line 440-443: it is not clear what you meant in this sentence. If these two animals had inconsistent paired detection between nasal swabs and lung samples please state it clear.

- Line 457: two of the aoudad were seropositive in the control group, why do you assume transmission did not occurred earlier?

- Line 478: this is a widely used assumption and true if you assume that your rt-PCR performed correctly, and you had no inhibitors. This sentence sounds awkward, I suggest rewriting it. In my opinion, the infective dose of M. ovipneumoniae you used in each group was probably different, because, if I understood well, you cultivated the inoculum in a medium in the “Movi” group but not in the “Wash” group.

- Line 487: in what group?

- Line 490-491: more efficient for what? Maybe M. ovipneumoniae was not present in the lungs at euthanasia but had earlier pathogenic roles in the lungs. Where the samples collected at the edge of the lesions? These results have previously been reported in natural outbreaks in bighorn sheep (lack of detection in the lungs in some individuals). Have you quantified DNA before performing the rtPCR? DNA should be always quantified before performing any PCR method, and PCR methods could be even adapted to these DNA quantifications. The author should know if DNA was correctly extracted from the lung tissues. Please clarify.

- Line 494-496: I do not understand this sentence, I guess because I still had not understood what methods you used to estimate the pre-inoculation native lung microbiota. Please review all the article for being clearer, especially on the methods.

- Line 499: I do not remember seeing any result on lkt-A detection in lungs. Please review and make it clearer if necessary.

- Line 521: the control-contact group also had lung lesions? Did you clearly mention these results? I do not remember.

- Line 526: you found lesions by histopathology on control animals but no consolidation of cranioventral lobes?

- Line 532: molecular detection pf BRSV on lung samples should be performed.

- Line 537-738: this sentence is vague and not supported by your experimental results. In my opinion, your study only demonstrate that Aoudad are susceptible to develop respiratory disease from sheep nasal washings and a nasal culture from bighorn, both being confirm for the presence of M. ovipneumoniae, but the etiological agents involved are not clearly determined. For what I understood, your inoculums may contain multiple bacteria or viruses, including different mycoplasma species, and at unknown concentrations.

- Line 538-540: this sentence should be corrected according to my previous comments. You have demonstrated the involvement of M. ovipneumoniae and Pasteurellaceae in the disease process, but you may have inoculated other agents as well.

- Line 540: I understood your contact-control group did not show clinical signs? The infective does can also influence the course of the disease, but you don’t know the dose you used. I also understood you did not exclude the presence of other pathogens in the inoculum used in the “Movi” group.

6. PLOS authors have the option to publish the peer review history of their article (what does this mean?). If published, this will include your full peer review and any attached files.

Reviewer #1: No

Reviewer #2: No

Reviewer #3: No

---

## [Author Response · Author response to Decision Letter 0]

19 Sep 2023

Revisions Requested by the Editor, Dr. Serrano

• “On the other hand, I think it would be worthy to include a little paragraph about the potential impact of Aoudad on native ungulate populations as is happening in southeast Spain (Cassinello, et al. 2004. Range expansion of an exotic ungulate (Ammotragus lervia) in southern Spain: ecological and conservation concerns. Biodiversity and Conservation 13, 851–866) with aoudad populations acting as reservoirs for sarcoptic mange (González-Candela et al. Population effects of sarcoptic mange in Barbary sheep from Sierra Espuña Regional Park, Spain. J. Wild. Manage. 40(3): 456-465)”.

o I have addressed the revisions by adding the citations and details suggested: Lines 79 to 84 of the revised document

• “Regarding the blood analysis, it keeps unclear to me whether you got blood samples for haematology at each sampling point or just at beginning and end of the study. Please, explain this. If you have got blood at each sampling point, I would recommend analysing the dynamic of your blood parameters. In any case, there is very little information about this haematological section neither in the methods nor in the results section.”

o Please find this addressed in the methods section. I also added a supplementary figure investigating the dynamics of blood parameters

• “Please, provide references for tooth eruption patterns. Ungulates are generally classified as Kids, juveniles, yearlings sub-adults (also prime-age) and adults”

o I have added two references and clarification on aging aoudad in the methods section: Lines 128 and 129

Revisions Requested by Reviewer #1:

• “The wild aoudad likely were already exposed to M. ovipneumoniae, as the diagnostic testing done at capture used a metagenomic method of unreported sensitivity, which was unable to be completed on half of the captured aoudad.”

o It was to our surprise that 2 of the aoudad arrived to the study site with serological titers to M. ovipneumoniae. It was the authors’ hope that this finding would further solidify the notion that aoudad are competent hosts for M. ovipneumoniae. While we cannot confirm the source of the 2 exposure events prior to capturing animals and bringing them to the study site, this novel finding suggests that intraspecific transmission is likely. The distribution of free-ranging aoudad does not overlap with any known domestic sheep and goat operations for approximately 100 miles. A lack of a known reservoir for M. ovipneumoniae in the habitats used by the aoudad causes us to believe this represented intraspecific transmission. The index source of initial exposure is still unknown for aoudad in Texas, however. 

• “These findings confirm that aoudad are competent hosts of M. ovipneumoniae, but because infection was documented prior to the study challenge and because no strain typing data is presented, the authors’ claim that transmission occurred among the aoudad in this study, although not unlikely, is not directly supported.”

o The idea that transmission occurred is most strongly supported by the fact that aoudad in the control group developed antibodies (seroconverted) despite never having been inoculated. While it is possible that 1 or more control AD were exposed to M. ovipneumoniae and we failed to detect it by serology (upon capture) or by qPCR on Jan 31st (inoculation day/ day 0 for Movi group AD) the fact that they may have already been carriers does not weaken the evidence for transmission in this study. In fact, if any aoudad in any study group were M. ovipneumoniae carriers before any inoculations occurred, this strengthens the argument that transmission can occur in aoudad naturally. If no aoudad were carriers prior to inoculation, then it is certainly possible that transmission to the control AD only occurred because the inoculated AD groups got an artificially high M. ovipneumoniae dose. Regardless of when or how control aoudad were exposed, the most parsimonious explanation for our combined results is explained by transmission. 

• “I am a little concerned that the control group was excluded from the clinical signs comparison. The authors indicated that no clinical signs were observed in the control group, but Table S3 does identify at least some clinical signs in two individuals in this group.”

o This has been addressed by clarifying the comparisons in the methods section (Lines 258-261) and by modifying Figure 1 to include cumulative mean clinical scores of control aoudad.

• “The use of MLST to identify strain types is described briefly in the M&M but no data are presented in the Results. The discussion indicates that ‘non-convergence’ resulted in the inability of this test to be used. I cannot find any of the MLST data in the supplemental tables either.”

o This has been addressed by adding pertinent details in lines 338-351 and by the addition of a multi-locus sequencing results appendix

• “Kindly provide more information about the 7 animals that died or were euthanized prior to the study, and these may provide insight into the disease and respiratory infections of the wild herd. Were they sampled for microbiologic testing? Were lung lesions observed? What was the trigger for the 3 euthanasias?”

o This was addressed in the methods section lines 118-128 and the results section lines 270-281. Specifically, we added details about the triggers of the euthanasia events and the other mortality events here. To address comments made by other reviewers, the details on mortality events occurring before data were collected have been removed so that only animals providing data used in this study are reflected. We collected samples for microbiological testing for all individuals reflected in the revised version of the manuscript. Details on the patterns of lung lesions are outlined in the results section and in further detail in Supplementary Table 1. 

• “Kindly provide more information about the challenges. In line 139-144 describing the origin of the challenge inoculum, no data is provide to confirm the growth of M. ovipneumoniae from the source animals. The dose and method of challenge inoculation of the first group is not described. In line 144-147, were the domestic sheep sources of the nasal washes re-tested at the time of collection to confirm the continued presence of M. ovipneumoniae? Were the challenge washes from individual domestic sheep or were they pooled prior to challenging the aoudad?”

o These questions have been addressed and clarified in lines 171 to 202 of the revised manuscript.

• “-Regarding the microbiologic testing during the project: Has the cELISA assay performance been validated in aoudad? I know it has been found to not work at all well in another Caprinae species, domestic goats. Can you clarify why it was only conducted on samples beginning 14 or 21 d post-challenge? Were no blood samples available from the time of capture or challenge? Can you explain why the attempt at MLST was done on the ‘last nasal swab’ sample, rather than on the DNA extract the lab had obtained from the earlier PCR detection?”

o The cELISA has not been validated for use in aoudad. However, in the absence of suitable secondary antibodies available for commercial purchase, we elected to use WADDL. I added clarification on the blood collection events to lines 204-212 of the revised manuscript. MLST was conducted on the last available swab to permit the maximum number of possible transmission events distinguishable by our inoculation methods.

• “Regarding the 16S metagenome data acquired from the aoudad samples taken at the time of initial capture, were you able to access the raw sequence data from the company that performed the testing, so as to query the dataset directly for M. ovipneumoniae 16S sequences? If not, can you confirm their data analysis methods would have detected these veterinary pathogens, if present?”

o NexGen Diagnostics uses proprietary methods to determine the microbiome present in each sample. They do not provide raw sequences. I cannot confirm their data analysis methods, but this lab is certified under the Clinical Laboratory Improvement Amendments of 1988(CLIA 88) as qualified to perform high complexity clinical laboratory testing.

• “Were the lung specimens submitted to WADDL selected to contain medium – large airways, the site of Movipneumoniae infection?"

o The pathologists, Drs. Clontz and Reach were made aware of the experimental design and exposure histories for all animals involved in the study. Considering this, the pathologists collected all necessary lymphatic and pulmonary samples necessary to detect the pathogens of interest given their combined extensive experience in ruminant pathology

• “L206 – 246 This data presentation is very difficult to follow. In general, I suggest using Table 1 to show all detailed sub-group specific results, and use the text to simply identify group- or timepoints which differed significantly. The data in the table should be complete – that is, the data from timepoint 0 d should be added.”

o Please find this addressed in the results section

• “L214-217. If I am understanding this correctly, the reader has to compare the study wide tallies with the post inoculation tallies to deduce that Movi was detected in at least 4 aoudad in the Movi group prior to the day of inoculation. Is that correct?”

o Please find this clarified in the revised presentation of Table 1

• “L218-219 Please add this data to Table 1. I also think it would help readability to move this sentence to be the first in the Results section.”

o Please find these details in Table 1 of the revised manuscript

• “L219-221 Please explain a little more. Was the ‘re-analysis’ done by you or by the laboratory that ran the tests? Is anything known about what caused this discrepancy?”

o Please find this addressed in lines 311 to 319 of the revised manuscript

• “L229 Table 1: Please add Day 0 results to Table 1. Indicate the lack of rtPCR data for the capture time point, and the method actually used at this time. For the Capture column, using the total number of animals as the denominator rather than the numbers for which results were obtained would be more clear. For example, the Movi group would say 0/2 (5Ins). My thinking is that if insufficient samples were obtained from some animals, negative results should not be tabulated.”

o Please find these changes addressed in Table 1 of the revised manuscript

• “L236-250 Like the results for the ‘Novi’ groups, I suggest that the details for the ‘Wash’ and ‘Control’ groups be left in the table, and only the significant effects detected be presented in the text”

o Please find these changes addressed throughout the results section

• “L242 Should still cite Table 1, not Table 2.”

o Corrected

• “L252-261 Like the PCR results, I suggest leaving the details to Table 2, and use the text to only report the summary results (i.e., the final two sentences).”

o Please find this change reflected throughout the results section

• L274-313: Like above, please report the data in the table, and save the text for summary findings and significantcomparisons.L315-342: Same comment as previous.

o Please find this change reflected throughout the results section

• “L347-350: Why all the differing euthanasia dates unbalanced across treatment groups?”

o Please find the relevant details added to the results section lines 278 to 281

• “L353: define CMS. Also, please explain what data are added to make the scores ‘cumulative’”

o Please find details added for clarification in the methods section lines 254 to 256. The cumulative mean score is a temporally dynamic measure of clinical scores recorded for each animal each day. It is cumulative because the mean score for each animal each day is calculated longitudinally. It was calculated using the cummean (cumulative mean) function in program R. 

• “L367-431: Suggest deleting most of the detailed textual description. Perhaps just present Figure 3 then retain the final paragraph summarizing the patterns by experimental group. (If considered important, perhaps the details could be moved to an appendix?)”

o Please find the revision addressed. I have also added the details to a pathology appendix.

• “L433-436 Suggest adding the observation of pre-existing infection in aoudad at capture, indicating infection by natural processes.”

o Please find this addressed in the very beginning of the revised manuscript

• “L436: The data presented may be consistent with intraspecific transmission, but they are not confirmatory: it is difficult or impossible to rule out persistent infections fluctuating above and below the detection threshold for PCR. However, if the inoculum strain types of M. ovipneumoniae, obtained pre-challenge, had been detected in experimental groups that had not received those challenges, that would have been direct evidence of transmission.”

o I addressed this in lines 476 to 489 of the revised manuscript. I added statements on the sources of uncertainty surrounding our claims of transmission to indicate that it was the combination of several pieces of evidence that supported natural transmission was likely rather than stating that our results directly proved transmission occurred. 

• “L437-440: Unless lung specimens had been selected for presence of large ciliated airways, the negative results from some animals do not rule out lower respiratory M. ovipneumoniae infection.”

o The pathologists, Drs. Clontz and Reach were made aware of the experimental design and exposure histories for all animals involved in the study. Considering this, the pathologists collected all necessary lymphatic and pulmonary samples necessary to detect the pathogens of interest given their combined extensive experience in ruminant pathology

• “L448: This seroconversion data may be the first evaluation of the cELISA test in the aoudad species. Worth mentioning?”

o I added in a statement indicating that this was the first published investigation of the cELISA in aoudad

• “L461: There is no multi-locus sequence data presented in the Results. Also, define ‘non-convergence’.”

o I added MLST data to lines 338 to 344 of the results section in the revised manuscript. Additionally, details on the laboratory findings are available in the MLST appendix 

• “L464: “…to confirm that M. ovipneumoniae is transmitted among aoudad” as I believe the data in this study is suggestive but not confirmatory.”

o Our statements were modified to include the sources of uncertainty surrounding our claims of transmission to indicate that it was the combination of several pieces of evidence that supported natural transmission was likely rather than stating that our results directly proved transmission occurred. 

• “L468: What possible sources of M. ovipneumoniae infections detected immediately pre-challenge exist, other than naturally occurring infections?”

o The serological status of all aoudad was negative prior to inoculation. Additionally, the fact that inocula were created shortly before capturing and collecting samples from aoudad on inoculation day (day 0) presented a contamination risk. We added statements indicating this possibility in lines 527-531.

• “L470-471: Multi-locus sequence typing would most likely be able to discriminate between these sources, but no data is presented in Results.”

o Please find MLST data presented in lines 338 to 344 and in the MLST appendix

• “L478-491: This paragraph is largely speculative and I suggest deleting it, to shorten a very long discussion.”

o Statements on Ct values, the relative bacterial load carried by different treatment groups, and DNA extraction efficiency were removed as requested. 

• “L499-501: I could not find any lktA detection data in the Results.”

o Please find this data added the lines 392 and 393 of the results section

• “L506-9 I could not find any data showing that the Wash group was exposed to leukotoxigenic Pasteurellaceae in the challenges.”

o I clarified this statement in lines 191 and 192 of the methods section

• “L526-528: The lab (WADDL) that runs this test cautions that the test is validated at the group level, not in individuals, because individual non-reactors have been observed in all validated subject species.”

o I addressed this in lines 501 to 510 of the revised manuscript

Revisions Requested by Reviewer #2

• “Why the different periods of time for acclimitization between the groups ?”

o Please find the relevant details added to the results section lines 278 to 281

• “The groups sizes are quite different – the transmission group would be very important as the authors are trying to make the point that auodad are a risk to bighorn sheep but it was only 3 individuals -why?”

o This was addressed in the methods section lines 118-128 and the results section lines 270-281. Specifically, we added details about the triggers of the euthanasia events and the other mortality events here. For our purposes, this study was the very initial step to determining whether aoudad were even capable of harboring and transmitting important pathogens. 

• “Why was ivermectin given on day 45 and not upon initial capture? How many ofthe animals had parasites upon deworming, and could this have affected the results (knowing that micro and macroparasites interact through the immune system)?”

o A detailed explanation is offered on lines 163 to 167 of the revised manuscript. While we understand that altering the microbiome and nemabiome of study animals may have altered study results, this was done for animal welfare. Additionally, we believe the overall effect to be quite minimal because the same ivermectin treatment was applied to all animals in the study equivalently. 

• “My main concern is that the title and abstract discuss the potential for transmission, however for M Ovi the only diagnostic reported is PCR, which includes live and dead bacterial signal. How do they know if the animals are able to transmit live pathogen? i.e. how can they assess risk to sympatric bighorn. The authors showed conclusively that auodad can get infected but I would argue the evidence for transmission is quite weak (1/3 sheep became PCR positive in the contact group)? Or are they arguing that because they took nasal washes from infected sheep and then moved those into susceptible sheep that that is a proxy for transmission potential? I assume the dose received in a nasal wash is much higher than in a normal sheep to sheep interaction? Some careful thought and discussion should be given to this potential.”

o Our main support for the ability of aoudad to transmit M. ovipneumoniae intraspecifically comes from the observed seroconversion of control aoudad from seronegative upon initial capture to seropositive (2/3) at the end of the study. We believe the qPCR finding bolsters our claim of transmission. While qPCR can detect live or dead pathogens, M. ovipneumoniae is biologically and antigenically fastidious, and we would not expect animals to seroconvert upon exposure to aerosolized dead bacteria (if such bacteria could even be transmitted via aerosolization once the cell membrane has lysed, since no cell wall is present in this species). While we were unable to quantify the doses given to each inoculated group, we expect that natural host-pathogen interactions within inoculated aoudad likely reached a biologically plausible and relevant equilibrium that reflects possible transmission contexts that could be experienced in free-ranging aoudad. That is, once the microbes from the inoculum were inoculated into the Movi and Wash groups, they likely shed a biologically plausible dose of M. ovipneumoniae that free-ranging individuals may encounter with intraspecific exposures. 

Revisions Requested by Reviewer #3

• “One of the main concerns is that the study failed to demonstrate if the animals incorporated in the study were free of M. ovipneumoinae before the experimental challenge.”

o The authors acquiesce that the 2 serologically positive aoudad captured and used in this study were analytically problematic from an experimental perspective. However, it is our belief that finding 2 free-ranging aoudad with serological exposure to M. ovipneumoniae supports the ultimate objective of the study: To determine whether aoudad are competent hosts that can transmit respiratory pathogens important for bighorn sheep management. Regardless of whether our results were primarily driven by inoculation or by natural transmission among aoudad before the study, our results demonstrated the biological capacity of aoudad to shed, transmit, and harbor M. ovipneumoniae. Admittedly, the presence of M. ovipneumoniae prior to the study could certainly impact the clinical scores and pathology observed in this study. To account for this, we describe the uncertainty surrounding the clinical responses of aoudad to these pathogens in lines 556 to 562 of the revised manuscript

• “In the methods they mention they used a next-generation sequencing method, without further specifications, to discard the presence of M. ovipneumoinae in nasal swabs. However, the method failed in about half of the animals. Did the authors tried to different dilutions to avoid inhibitors? Did you try a specific rt-PCR for Movi as used in subsequent samples? This should be a more sensitive method as typically amplifies shorter DNA fragments, and it seems you did not have inhibitions in subsequent sampling. Why you do not show the results of the 16SrRNA sequencing to describe the nasal microbiome prior the challenge? The status of animals before the challenge is critical and it is not well understood in the article, the authors should confirm either by serology (at least) or molecular detection the free status of all animals and clearly state it in the manuscript. Currently it is not known if the animals that with invalid results in nasal swabs were at least seronegative.”

o Next generation sequencing was used to pool resources with our collaborator, who wanted to investigate nasal microbiome changes in response to capture and transit. The next generation sequencing was performed using proprietary methods by MicroGen Diagnsotics, which is certified under the Clinical Laboratory Improvement Amendments of 1988(CLIA 88) as qualified to perform high complexity clinical laboratory testing. We do not know whether they performed proper dilutions, but it is reasonable to assume that this high quality diagnostic performed the steps necessary to accurately ascertain the microbiome. We did not show the sequences of the Next generation sequencing protocol because the laboratory only reports the species composition in their results. Further, this study’s primary objective was to determine the ability of aoudad to harbor, shed, and transmit M. ovipneumoniae and was not focused on assessing the full spectrum of microbiome changes and specific agents that may have been associated with any pathology developed. While we cannot definitively state the initial molecular status of all animals upon capture, we determine that only 2 aoudad were previously exposed to M. ovipneumoniae as illustrated by the presence of detectable M. ovipneumoniae-specific antibody titers. Please find the details on serological status of aoudad upon capture reflected in Table 2 and recapitulated in the results section-lines 345 to 347.

• “Another concern is that, according to the methods described, the authors did not test what was present in the inoculums used (nasal wash and broth culture from a nasal swab), neither the dose of M. ovipneumonniae, the presence of other bacteria in the culture-broth inoculum or the lkt-A virulence gene as the title of the article claims.”

o We added details to the methods lines 173 to 176 of the revised manuscript to indicate that the Movi group inoculum was derived from swabs that were culture and qPCR positive. The culture techniques by Wyoming State Veterinary Laboratory isolated as indicated in lines 178 and 179 of the revised manuscript. We describe the limitations of the wash group inoculum in lines 182 to 192 of the revised manuscript. However, we support our methods by outlining the frequent carriage of leukotoxigenic Pasteurelleaceae by domestic sheep. Then leukotoxicity of the wash group inoculum was demonstrated not only by detection of the leukotoxin-A gene only in wash group aoudad, but oat cell formation observed in histopathology further supported the involvement of leukotoxin-A in the clinical dynamics of wash group aoudad. 

• “Both the title and the methods mention the detection of lkt-A virulence gene, but I have not found these results in the manuscript.”

o Leukotoxin data has been added to lines 392 and 393 of the revised manuscript

• “the control-contact group was not sampled until the end of the study period”

o This detail has been added to lines 162 and 163 of the revised manuscript

• “an antiparasitic treatment was applied in the middle of the experimental infection and it is unknown how differences on parasitism could have affected the response of the challenges”

o A detailed explanation is offered on lines 163 to 167 of the revised manuscript. While we understand that altering the microbiome and nemabiome of study animals may have altered study results, this was done for animal welfare. Additionally, we believe the overall effect to be quite minimal because the same ivermectin treatment was applied to all animals in the study equivalently. 

• - Line 46-47: what is an exotic or native interaction?

o Corrected and described in line 64 of the revised manuscript

• - Line 47-49: I am not a native English speaker, but verb tenses does not seem to match in this sentence.

o Please find this corrected in lines 64 to 66

• - Line 58-61: please add a reference to the statement.

o Completed

• - Line 78-80: please clarify if you refer to bighorn sheep in this sentence.

o Corrected in line 103

• - Line 80: this is based on targeted studies? Passive/scanning surveillance?

o This is based on active surveillance of bighorn sheep and aoudad

• - Line 80: prior to what?

o Please find this clarified in lines 101 to 106

• - Line 83-84: in my opinion you did not test the susceptibility to M. ovipneumoniae with or without co-exposure to Pasteurellaceae, because you used an inoculum with unknown infectious agents and doses. Please review your objectives in relation to the study design and methods.

o Please find the title and lines 108 to 110 updated to reflect this suggestion

• - Line 86: transmission or shedding?

o Corrected in lines 110 t0 112 of the revised manuscript

• - Line 101: please define the reasons for euthanasia.

o This was addressed in the methods section lines 118-128 and the results section lines 270-281. Specifically, we added details about the triggers of the euthanasia events and the other mortality events here. To address comments made by other reviewers, the details on mortality events occurring before data were collected have been removed so that only animals providing data used in this study are reflected. We collected samples for microbiological testing for all individuals reflected in the revised version of the manuscript. Details on the patterns of lung lesions are outlined in the results section and in further detail in Supplementary Table 1. 

• - Line 107: “control-contact” may define better this experimental group, as this is not exactly a control group.

o This has been corrected throughout the manuscript

• - Line 112: acclimatation period? But inoculations were already done?

o Clarification added to lines 134 to 140 in the revised manuscript

• - Line 114: please explain the methods to estimate the body condition daily. Were these methods the same as the ones used on capture events?

o Clarification added to lines 140 to 143 of the revised manuscript

• - Line 121: please correct to “captures”. I think “initial” does not add any significant information or clarification, but it raises the question if the capture methods in Carrizo mountains have changed over the study.

o Corrected

• - Line 124-125: what do you mean by thereafter? Every 30-45 minutes? Please review the sentence, it is not clear. If there were consecutives doses and a defined period for acclimatation to captivity, please specify in the sentence.

o Clarified on line 150 of the revised manuscript

• - Line 126-127: The way it is written is confusing. Instead of linking the methods with a period of time, which has no logical connection (“after acclimatation”), please mention the purpose of those chemical immobilizations. At this point, the reader does not know why you are doing these immobilizations.

o Clarification added to line 152 of the revised manuscript

• - Line 127: How was stress handled during those immobilizations? And how was chemical immobilization applied? Were the animals physically restrained before, on a crate, with guns, etc… please provide more details on the methods specially focusing on how stress was managed.

o Details added in lines 152 to 156 and 168 to 169 in the revised manuscript

• - Line 134: please clarify if ivermectin was applied to all experimental groups. Why was the treatment not applied during the acclimatation period and before the infection?

o A detailed explanation is provided in lines 163 to 169

• - Line 141: I believe the modified tryptic soy broth is not a specific medium for the growth of mycoplasmas. Why did you choose this medium? How did you discarded the growth of other bacteria than M. ovipneumoniae.

o Details added to lines 172 to 182. This is the culture medium used frequently and successfully by Wyoming State Veterinary Laboratory

• - Line 139-144: How the inoculum was quantified for M. ovipneumoniae? Was the culture broth tested for other bacteria? Where other mycoplasmas discarded? Please clarify.

o Details added to lines 173 to 176 of the revised manuscript

• - Line 145: please make sure that you have explained what is “DS”. I am not sure if this was mentioned before.

o Corrected

• - Line 147: qPCR and rt-PCR are abbreviations to the same method, either referring to the concept of “quantitative” or “real-time”. You previously used the standardized abbreviation of qPCR, if you also refer here to the same method pleaseuse the same abbreviation. It is weird to use different abbreviations for the same method, even if the primers and probes used are different.

o Corrected throughout the manuscript

• - Line 149: why the Wash inoculum was also administrated into conjunctival sacs? Is this a demonstrated infection route for M. ovipneumoniae? Where the authors also interested on demonstrating the pathogenic effect of other bacteria on ocular and conjunctival tissues? Please explain.

o Reference number 24 in the revised manuscript reflects the expertise of Thomas Besser in Bighorn sheep pneumonia etiology. We adopted their inoculation protocol. The objective of the protocol (as explained to me by Dr. Besser and his laboratory staff) is to expose the animal to all possible routes of infection through available ciliated mucosa of the head. While we know the disease is respiratory in nature, we do not actually know which mucosa is first colonized. 

• - Line 150-151: Negative to what sample? It is not explained and even if it is explained later in the text it should be clarified in this sentence. What is the rationale for performing the re-inoculations? Even if the animals presented respiratory clinical signs were re-inoculated?

o Details and clarification added to lines 197 to 202. 

• - Line 160-161: why antibodies against M. ovipneumoniae were not tested at day 0 and consecutive days to test for serconversion? This information must be included as it can affect the results of the experimental infection.

o Details and clarification added to lines 209 to 212 of the revised manuscript

• - Line 163: from what sampling?

o This is describing the method for all sampling events. Details are added in lines 214 to 218 of the revised manuscript

• - Line 163-167: The Muller-Hintor medium is non-specific for Pasteurellaceae, how were the colonies selected for further identification? How were the isolates identified? Please describe properly the methods used.

o This was conducted under standard procedures at Texas Veterinary Medical Diagnostic Laboratory. Professional microbiologists isolated colonies and confirmed their speciation by matrix assisted laser desorption ionization-time of flight mass spectrometry. Details provided in lines 218 to 220 of the revised manuscript

• - Line 167: please explain properly the experimental design, including sampling days for each group, pathogen, etc…

o Please find the requested changes on lines 154 to 156 of the revised manuscript

• - Line 170-171: please provide the reference for the PCR used and the threshold used for determining the sample’s Ct.

o Details added to lines 220 to 224 in the revised manuscript

• - Line 171-172: MLST is usually performed from pure cultures, not field samples. If there is more than one strain the approach is useless. Please explain better the methods.

o Details added to lines 227 to 230 of the revised manuscript

• Line 173: please describe the details of the next-generation sequencing, this is an umbrella term for different methods. Was the 16SrRNA gene full-length sequenced? Overall, it is surprising that the authors used a next generation sequencing just to discard the presence of M. ovipineumoniae in nasal swabs. Why the authors did not use the results on the detection of other bacteria that could be involved in respiratory disease (Pasteurellaceae, Moraxella spp., etc.)?

o Clarification added to lines 231 to 235 of the revised manuscript. The details of the technique are proprietary at Microgen Diagnostics in Lubbock, Texas. The scope of the study was not focused on describing the full array of potential pathogens involved in any respiratory disease that developed due to inoculation or other experimental activities. The objective of our study was to determine the dynamics of M. ovipneumoniae and Pasteurellaceae in aoudad. 16S was performed at capture as a collaborative effort to pool resources and address a separate study on microbiome changes associated with capture and translocation

• - Line 179: what sections of lungs? From all the lung lobes? Only from lesions?

o The pathologists, Drs. Clontz and Reach were made aware of the experimental design and exposure histories for all animals involved in the study. Considering this, the pathologists collected all necessary lymphatic and pulmonary samples necessary to detect the pathogens of interest given their combined extensive experience in ruminant pathology

o 

• - Line 182: please specify which tissues were cultured.

o Clarified in lines 242 and 243 of the revised manuscript

• - Line 187-189: this is not clear. What time periods? Do you mean sampling days?

o Clarified in lines 247 to 250 of the revised manuscript

• - Line 189: What groups were compared? Sex? Age? You need to be always specific. If the you compared the “Movi”,“Wash” and Control group you need to mention it.

o Clarified in lines 249 and 250 of the revised manuscript

• - Line 191: how many times was the clinical scores recorded per individual and day to calculate the mean?

o Clarified in lines 254 and 256 of the revised manuscript

• - Line 194: How were compared?

o Clarified in lines 253 of the revised manuscript

• - Line 202-203: this sentence is not necessary and weird given that the authors did not mention anything about degrees of freedom before.

o Please find this statement removed

• - Line 208: please specify to what the samples were possible.

o Please find this clarified in lines 208 to 209 of the revised manuscript

• - Line 210-211: by what method? If it was by metagenomics?

o Please find this addressed in lines 300 to 304 in the revised manuscript

• Line 212: I understood there were two treatment groups and a contact control?

o This specific information has been removed from the text and is instead represented in Table 1 in the revised manuscript; per other revision suggestions

• - Line 213: positive to what? You need to be specific in each sentence to avoid any doubt. Please correct.

o Please find this clarified in lines in each paragraph of the M. ovipneumoniae subsection of the results section of the revised manuscript

• - Line 214: what is the different between swabs collected post-infection and study-wide? This is confusing. I would assume that study wise also includes the samples collected at field captures.

o This specific information has been removed from the text and is instead represented in Table 1 in the revised manuscript; per other revision suggestions

• - Line 206-217: this paragraph is impossible to follow, please rewrite to be clearer with the methods.

o Please find clarifications made to lines 298 to 310 of the revised manuscript

• - Line 218: does this mean that the “Movi” group already had Mycoplamsa ovipneumoniae before inoculation? You mention you had 7 positives but I believe there were 6 animals in that experimental group? Please clarify.

o Clarifications to animal numbers have been addressed per previous comments. We mention later in line 529 of the discussions section that handling and creating the inoculum solutions prior to collecting day 0 samples may have presented a sample contamination risk.

• - Line 222: seven days post-infection again? You need to use a clearer explanation of the methods.

o Please find this clarified in lines 313 to 319 in the revised manuscript

• - Line 223: what analyses? And why you mention it here?

o Please find this clarified in lines 319 to 321 in the revised manuscript

• - Line 224: please make sure you correctly explain the experimental design in the methods, including the sampling days per experimental group etc.

o This specific information has been clarified and/or removed from the text and is instead represented in Table 1 in the revised manuscript; per other revision suggestions

• - Line 229: in the results section you have 14 animals divided in three experimental groups, but in the table there are 15animals at capture. I assume you had casualties before the inoculation to explain this mismatch, but in this case, I suggest removing them from the table to keep consistency on the number of animals you incorporated in the experimental study. The results of animals not incorporated in the experimental infection you can mention them separately in the text.

o Please find these details clarified in lines 119 to 128 and lines 270 to 281.

• Line 229: In the methods of the article the detection of M. ovipneumoniae at capture is perfromed by amplifying the16SrRNA using a next-generation sequencing, but here you mention that you detected it by rtPCR. Please clarify, or correct.

o Please find this corrected in Table 1

• - Line 230: I don’t think percentage is appropriate for such low numbers. I suggest to include the proportions as the main result, and the authors can decide whether they still want to keep the percentage in brackets. Line 230: the format of the table in the pdf I have is awkward and not the same in all columns and rows. Perhaps this is an artifact in the generation of the pdf, but please make sure you comply with the journal guidelines for tables. Line 231: In my opinion, this should be included in title of the table, not as a footnote. Please describe “P.I.” and “NA” in the footnotes.

o Please find all these addressed in Table 1 of the revised manuscript

• Line 236: positive to what? What is insufficient? Is this information any relevant? Why does not match with the total P.I. column in table 1? Even adding the swabs at captures does not sum up to 30.

o This was addressed by adding in data to account for all sampling events. Additionally, per other suggestions, most descriptive data has been left in the tables with only significant details and relationships described in the text. 

• - Line 239-240: you had positive samples to M. ovipneumoniae before the inoculation? Please clarify?

o Yes, this is now reflected in Table 1 of the revised manuscript and is discussed in lines 311 and 321 of the results section and line 529 of the discussion section of the revised manuscript

• - Line 240-243: this information is redundant with the table, please remove and reference to Table 1.

o Please find this corrected

• - Line 242: if I understood well this information refers to Table 1, not Table 2.

o Yes, please find this corrected

• Line 243-244: in line 236 you state that throughout the study you detected 50% of positive samples in the Wash group. What is the difference between “throughout” and “overall”? You should use clear terms and consistent throughout the article.

o This was addressed by removing redundant information present in Table 1

• - Line 245: “after the Movi and Wash group were inoculated” is not a precise point of time, please be specific.

o Addressed by adding specific reference days

• - Line 250-251: In my opinion, this test statistic is not necessary.

o Please find this removed

• - Line 267: Please see my comments on Table 1 because many still apply in this table.

o Please find these changes applied to all tables in the manuscript

• - Line 280: why mixed infections could not be identified? I understood that you did microbiological cultures for obtaining pure cultures. Furthermore, I believe I have not read the methods on how you identified or detected Pasteurellaceae. Please review all methods and make sure that are all clear.

o Please find these details in lines 214 to 220 and lines 373 to 376

• Line 281: how swab samples pre inoculation sum up to 27 if you have 15 individuals divided in three experimental groups? I can not understand either by looking at the numbers of the table. Please explain or make it clearer in the text.

o This was organically addressed when revising the suggested revision for Line 229. Additionally, excess detail was removed from the text and the Tables were modified to include all data

• - Line 285: please review my previous comments on Table 1 because some also apply to Table 3.

o Please find these changes applied to all tables in the manuscript

• - Line 293-298: Please remove redundant information that already appears in table 3.

o Please find these changes made

• - Line 304: please change “detection” by “isolation”.

o Done

• - Line 305-307: what experimental groups?

o Please find this addressed in lines 385 to 388 of the revised manuscript

• - Line 310: more frequently than what other group?

o Please find these changes made to lines 389 to 391 of the revised manuscript

• - Line 313: in the materials and methods you mention that you did a PCR for the presence of the lkt-A gene, but you do not provide these results. Can you clarify this?

o Please find these results in lines 392 and 393 of the revised manuscript

• - Line 311-313: the culture methods used are not very sensitive for the detection of diverse Pasteurellaceae, especially considering culture inhibitions among them (see Kugadas et al., 2014.Veterinary Microbiology, 174:1-2, pp 155-162.). In my opinion, it has not much sense to compare specific differences on the isolation of different Pasteurellaceae for specific sampling days.

o Please find temporal species specific comparisons removed as suggested

• - Line 343: I suggest moving this section to the beginning of the results because it can help to understand why the animals sampled throughout the study vary.

o Please find the revised version of the manuscript reorganized as suggested

• - Line 344: you also inoculated the conjunctival sac; no ocular lesions were detected over the course of the experimental infection?

o This is addressed in the results section and in the pathology appendices

• Line 348 and line 349: please explain the reasons why the animals 1 and 13 were euthanized.

o Please find these details in lines 270 to 281 and lines 121 to 127

• - Line 360: why the CMS drop suddenly in the Wash group at around 38 dpi? Please explain briefly in the table footnote or title, because this trend is not possible to understand without explanations

o Please find this detail added to the figure 1 caption

• - Line 379: what about pathological lesions in the lungs of the control-contact group?

o Please find these details in lines 464 to 470 of the revised manuscript

• - Line 398: did you perform cultures from the caseous lymphadenitis or other lesions observed a part of the lungs?

o Yes, lungs were subjected to culture as described in lines 236 to 241 of the revised manuscript. We added details on lung culture results in lines 391 to 396 of the revised manuscript. Finally, these details were added to Supplementary table 1. 

• - Line 436: the transmission between aoudad groups probably happened before 53 d.p.i, as the animals had seroconverted, therefore “for at least” does not sound completely correct here. Please review.

o Please find this addressed in line 477 of the revised manuscript

• - Line 440-443: it is not clear what you meant in this sentence. If these two animals had inconsistent paired detection between nasal swabs and lung samples please state it clear.

o Please find this clarified in lines 491 to 496 of the revised manuscript

• - Line 457: two of the aoudad were seropositive in the control group, why do you assume transmission did not occurred earlier?

o I addressed this by removing the assumption about when transmission occurred and instead discuss the finding that seroconversion was incomplete in the control group. This detail can be found in lines 508 to 515.

• - Line 478: this is a widely used assumption and true if you assume that your rt-PCR performed correctly, and you had no inhibitors. This sentence sounds awkward, I suggest rewriting it. In my opinion, the infective dose of M. ovipneumoniae you used in each group was probably different, because, if I understood well, you cultivated the inoculum in a medium in the “Movi” group but not in the “Wash” group.

o I removed details on CT values and relative bacterial M. ovipneumoniae loads between groups

• - Line 487: in what group?

o These comments were removed to address other reviewer suggestions

• - Line 490-491: more efficient for what? Maybe M. ovipneumoniae was not present in the lungs at euthanasia but had earlier pathogenic roles in the lungs. Where the samples collected at the edge of the lesions? These results have previously been reported in natural outbreaks in bighorn sheep (lack of detection in the lungs in some individuals). Have you quantified DNA before performing the rtPCR? DNA should be always quantified before performing any PCR method, and PCR methods could be even adapted to these DNA quantifications. The author should know if DNA was correctly extracted from the lung tissues. Please clarify.

o We removed statements on DNA extraction efficiency. Samples were collected at the edge of lesions by a board-certified veterinary pathologist (Dr. Raquel Rech). Details of DNA extraction and qPCR are unavailable, as they were performed at fee-for-service veterinary diagnostic laboratory. I am confident that the Washington Animal Disease Diagnostic Laboratory uses consistent and accurate measures of DNA in all qPCR protocols. 

• - Line 494-496: I do not understand this sentence, I guess because I still had not understood what methods you used to estimate the pre-inoculation native lung microbiota. Please review all the article for being clearer, especially on the methods.

o The inoculation methods section has been revised to describe the differences between the Movi and Wash group inoculum. Additionally, more details on the culture methods is provided.

• Line 499: I do not remember seeing any result on lkt-A detection in lungs. Please review and make it clearer if necessary.

o Please find these results provided in lines 393 and 394 of the revised manuscript

• - Line 521: the control-contact group also had lung lesions? Did you clearly mention these results? I do not remember.

o These details can be found in lines 427 to 452 of the revised manuscript

• - Line 526: you found lesions by histopathology on control animals but no consolidation of cranioventral lobes?

o That is correct, please find the details in lines 427 to 452 of the revised manuscript

• - Line 532: molecular detection pf BRSV on lung samples should be performed.

o Unfortunately, we are unable to oblige this suggestion, as Washington Animal Disease Diagnostic Laboratory destroyed these samples when they moved their laboratory to a new facility. 

• - Line 537-538: this sentence is vague and not supported by your experimental results. In my opinion, your study only demonstrate that Aoudad are susceptible to develop respiratory disease from sheep nasal washings and a nasal culture from bighorn, both being confirm for the presence of M. ovipneumoniae, but the etiological agents involved are not clearly determined. For what I understood, your inoculums may contain multiple bacteria or viruses, including different mycoplasma species, and at unknown concentrations. Line 538-540: this sentence should be corrected according to my previous comments. You have demonstrated the involvement of M. ovipneumoniae and Pasteurellaceae in the disease process, but you may have inoculated other agents as well.

o Please find the discussion section revised to account for the uncertainty of the pneumonia etiology and find clarifications on how the aoudad pathogen dynamics presented here support transmission risks to bighorn sheep occupying the same landscape

• - Line 540: I understood your contact-control group did not show clinical signs? The infective does can also influence the course of the disease, but you don’t know the dose you used. I also understood you did not exclude the presence of other pathogens in the inoculum used in the “Movi” group.

o While we were unable to quantify the doses given to each inoculated group, we expect that natural host-pathogen interactions within inoculated aoudad likely reached a biologically plausible and relevant equilibrium that reflects possible transmission contexts that could be experienced in free-ranging aoudad. That is, once the microbes from the inoculum were inoculated into the Movi and Wash groups, they likely shed a biologically plausible dose of M. ovipneumoniae that free-ranging individuals may encounter with intraspecific exposures.

---

## [Decision Letter · Decision Letter 1]

9 Nov 2023

Evaluating the transmission dynamics and host competency of aoudad (Ammotragus lervia) experimentally infected with Mycoplasma ovipneumoniae and leukotoxigenic Pasteurellaceae

PONE-D-23-16154R1

Dear Dr. Thomas,

We’re pleased to inform you that your manuscript has been judged scientifically suitable for publication and will be formally accepted for publication once it meets all outstanding technical requirements.

Kind regards,

Emmanuel Serrano, PhD

Academic Editor

PLOS ONE

Additional Editor Comments (optional):

Reviewers' comments:

Reviewer's Responses to Questions

**Comments to the Author**

1. If the authors have adequately addressed your comments raised in a previous round of review and you feel that this manuscript is now acceptable for publication, you may indicate that here to bypass the “Comments to the Author” section, enter your conflict of interest statement in the “Confidential to Editor” section, and submit your "Accept" recommendation.

Reviewer #1: (No Response)

Reviewer #2: All comments have been addressed

Reviewer #3: (No Response)

2. Is the manuscript technically sound, and do the data support the conclusions?

Reviewer #1: Partly

Reviewer #2: Yes

Reviewer #3: Partly

3. Has the statistical analysis been performed appropriately and rigorously? 

Reviewer #1: I Don't Know

Reviewer #2: Yes

Reviewer #3: Yes

4. Have the authors made all data underlying the findings in their manuscript fully available?

Reviewer #1: Yes

Reviewer #2: Yes

Reviewer #3: Yes

5. Is the manuscript presented in an intelligible fashion and written in standard English?

Reviewer #1: No

Reviewer #2: Yes

Reviewer #3: Yes

6. Review Comments to the Author

Reviewer #1: General comments: I still think this project is important and carries significant implications. I find their data on infection of auodad by M ovipneumoniae convincing, overall. I find their data on transmission much less so, and suggest it be presented as data consistent with, but not absolutely confirmatory of transmission (due to uncertainty about infection status at capture, inability to document strain types, etc.)

The paper is not at all tightly written and would benefit greatly by editing for directness and brevity. The Discussion in particular wanders, is redundant in places, and overall fails to present a clear picture of what the authors' data does and does not support.

L72-4 Citation needed for serologic evidence

L137 clinical scoring cannot evaluate pathogenesis, or potential for pathogen shedding.

L158-162 seems to discount the animal euthanized post-capture pre-study for haemonchosis

L167 These three refs provide three different sets of primers/probe for Movi. Were all really used?

L172-L177 In vitro culture of Movi has been associated with attenuation of virulence (Besser 2008)

L179-181 Provide citation. This statement contradicts my personal experience of variable shedding of Movi in adult DS

L181-2 The same single wash used in all four? Or single washes from four different DS used in four different BHS?

L183-185 evidence for multiple strains? Time between wash collection and inoculation, and conditions of storage if any?

L188-191 citation by reference number would be sufficient, as in the rest of the paper.

L305-308 confusing: Movi ‘consistently detected’ yet ‘both groups indeterminate throughout the study’?

L335-341 The available sequence data suggests that only a single Movi strain was present in the DS

L347-348 No sequences are reported in Appendix 1. If there are sequences they should be reported. If no sequences were obtained, this sentence should be edited to indicate that.

L349-355 Even in hosts where the cELiSA has been validated, WADDL report interpretive comment suggests limiting interpretation to groups (rather than individuals).

L385-387 OR values suggest the groups have been reversed.

L471 The discussion is overly long, redundant in places, and confusing in others. Suggest the authors identify the specific topics most important to discuss and clarify the discussion of those points, eliminating discussion of confusing or contradictory findings. However, it would be good to add a paragraph clearly listing and describing all the limitations of this study in one place.

L503-510 Interpreted on a group basis, consistent with transmission. Interpretation in individuals is discouraged in all species where the cELiSA test is validated due to individual variations. Suggest editing to remove attempt at individual animal level.

L511-514 Not surprising: animals were not sampled sufficiently to exclude transient infection. Seroconversion is not expected to be complete, based on results in species where the test has been validated.

L514-517 Confusing and unclear.

L501 – 522 Paragraph wanders. What is the intended point? I read it as saying ‘Some but not all evidence points to transmission among the study groups, but this could not be confirmed.’

L524 should also mention the totally unknown sensitivity and specificity of the 16S method used at capture, which render even results from animals with sufficient ‘sample rate’ difficult or impossible to interpret.

L574 - 576 Clinical score results (L279-288) suggest minimal disease in the control-contact group. I suggest that the authors explicitly consider the possibility that the frequent repeated captures of the Movi and Wash group animals may have directly contributed to their development of respiratory disease signs, independently of their (basically unknown) pathogen status.

L589 Having just carefully read the results and the discussion up to this point, it is not clear that the statement in this line is supported. Data has been presented both supporting and failing to support this statement. Suggest replacing here (and in the abstract) with a more tempered statement.

Reviewer #2: (No Response)

Reviewer #3: Review of PONE-D-23-16154R1

This study has major flaws in the experimental design and there is not much to do about it. I still think that it contains relevant information to its field, yet the article remains clumsy, very difficult to follow and some of the methods are still not well described. This is surprising considering that this article is signed by several co-authors that should have at least carefully reviewed the manuscript, and this should not be the role of a reviewer. Although the authors have addressed the specific comments from the previous review, the experimental design is scattered throughout the methods, and it is very difficult for a reader to understand the study without reading forth and back. I strongly suggest the authors to first describe the study design more carefully, including groups, sampling times, and what diagnostic methods were used in each sampling in relation to objectives, to later explain thoroughly each laboratory and analytical method. The use of further subheadings may help to organize the manuscript. Please move the methods of the inoculums to material and methods, not results. I also recommend removing the MLST to track transmissions as this method failed for that purpose due to a wrong experimental design, and only provides information on the sequence type used in the inoculum (if I understood well…). Accordingly, the study does not provide data on “transmission dynamics” as the title claim, and the experimental infections is not even designed for that purpose. The only transmission tested in the study is in the control group, which was only sampled at the end of the study, not allowing for investigations on dynamics. Furthermore, the control group is underrepresented with only three individuals included (vs 5 and 6 in other challenged groups) and neither the title nor discussion should mainly focus on transmissions. Instead, this study provides interesting data on shedding (no transmission), carriage and susceptibility of Aoudad to a polymicrobial respiratory disease that includes M. ovipneumoniae and Pasteurellaceae. Please pay careful attention in the wording, as molecular detection, or culturing in a broth without getting a pure culture cannot be called “isolate”. The discussion is clear and well-written.

Specific comments:

Lines 129-132: although I suggest keeping this information (animals IDs in relation to study groups, if euthanized, age, sex, etc), this information may be better to move to supplementary information, in case you are able to disclose most important results without it. Readers won’t be tracking animals’ IDs throughout the article.

Line 139: please rephrase “inoculation with isolated Mycoplasma ovipneumoniae”. To my understanding, you cultured a positive sample, without isolation. Furthermore, and unless the authors can justify, any other mycoplasma present in the swab sample would grow as well. Please rephrase accordingly. This is an important flaw of the study that should be stated and discussed properly.

Line 174: it is not clear why veterinarians recommended the treatment, please expose the reason upfront. It only becomes clear when reading other sentences or parts of the manuscript.

Line 187: the swabs were confirmed as positive by culture or by pPCR? The first sentence of the paragraph contradicts sentence in lines 187-188.

Line 197: exposed or infected? I assume you mean infected, please correct.

Line 230: what is this for? What group? What purpose? When? The manuscript remains clumsy and difficult to follow. You present methods without explaining the experimental design. I think it would be better to describe the experimental design first, including tests and sampling dates, and later describe the methods used.

Line 243: BHS nasal swabs from when? After inoculation? Please specify.

Line 246: natural transmission events within the study? Or previous? I think it would only demonstrate that AD were already infected with other M. ovipneumoniae strains prior to the challenges.

Line 326-328: this should be in methods when describing the experimental design.

Line 333-334: the swabs were insufficient? This sentence does not sound correct to me.

Line 328-330: is this information any relevant without specifying groups or times?

Line 338-341: it seems to me that these sentence are more related to next paragraph.

Line 354: how do you know this is a diagnostic discrepancy? Or it was a failure on the challenge? Please consider rephrasing it.

Line 355: “Because most of the individuals in Movi group…” please correct.

Line 356: please consider using the “colonize” word as a concept in previous sentences, I think these previous sentences were not very clear. I also think that it is not clear how you collected these time 0 swab samples, just after challenge or few hours later? Because if it was right away after the challenge you are just testing again the inoculum and is useless information.

Line 383: if I understood well, you did not isolate M. ovipneumoniae, you cultured a field sample without knowing what was there other than M. ovipneumoniae DNA. And what are these isolates? From what animals, groups or days do they come from (i.e. sample C32?)? Did you introduce the information about them before? Does it worth to provide a sample ID in the text? Because I do not see these sample Ids mentioned anywhere else. This is chaotic…

Lines 382-388: I think this information belongs more to material and methods to explain how you prepared your inoculums. Please organize your manuscript.

Line 382-388: MLST is usually performed from isolates, not from field samples. Chances of not getting conclusive results from field samples are high, and I would consider that this is a wrong study design. Mycoplasmas are not easy to culture, but not impossible, and M. ovipneumoniae is not even the worst mycoplasma to culture. Because your results are not conclusive on tracking transmission events, I would move this information to material and methods to just report the ST for each inoculum.

Line 347-348: do you mean that all AD were already positive to M. ovipneumoniae before the inoculation? Please review the sentence.

Line 452: correct p-value to three decimals, as it appears in other sections.

Line 455: do not abbreviate the genus if it has not been mentioned before.

Line 643: 8 negative individuals in the control group? The lesions are consistent with M. ovipneumoniae or with bacterial disease? Please rephrase.

Line 686-689: the authors really need to explain how these nasal samples on day 0 (same day of the challenge) were obtained.

Line 720: it was more common in nasal swabs? In lungs at the end of the study?

Line 768-769: plus BHS flora able to grow in the culture of the nasal swab samples from BHS (inoculum Movi group). Please reprase.

Line 779-780: I do not agree that this is something necessary, we already know BHS are susceptible, and transmission events in nature does not only depend on susceptibility, but also on effective interspecific contacts that are difficult to reproduce in experimental settings. I do agree that carriage, persistence/shedding in AD or epidemiological patterns are gaps of knowledge of interest.

7. PLOS authors have the option to publish the peer review history of their article (what does this mean?). If published, this will include your full peer review and any attached files.

Reviewer #1: No

Reviewer #2: No

Reviewer #3: No

---

## [Editor Report · Acceptance letter]

16 Nov 2023

PONE-D-23-16154R1 

Evaluating the transmission dynamics and host competency of aoudad (*Ammotragus lervia*) experimentally infected with *Mycoplasma ovipneumoniae* and leukotoxigenic *Pasteurellaceae*

Dear Dr. Thomas:

I'm pleased to inform you that your manuscript has been deemed suitable for publication in PLOS ONE. Congratulations! Your manuscript is now with our production department. 

Kind regards, 

on behalf of

Dr. Emmanuel Serrano 

Academic Editor

PLOS ONE